# Reliable Off-Policy Learning for Dosage Combinations

**Jonas Schweisthal, Dennis Frauen, Valentyn Melnychuk & Stefan Feuerriegel**
LMU Munich
Munich Center for Machine Learning
{jonas.schweisthal,frauen,melnychuk,feuerriegel}@lmu.de

## Abstract

Decision-making in personalized medicine such as cancer therapy or critical care must often make choices for dosage combinations, i.e., multiple continuous treatments. Existing work for this task has modeled the effect of multiple treatments *independently*, while estimating the *joint* effect has received little attention but comes with non-trivial challenges. In this paper, we propose a novel method for reliable off-policy learning for dosage combinations. Our method proceeds along three steps: (1) We develop a tailored neural network that estimates the individualized dose-response function while accounting for the joint effect of multiple dependent dosages. (2) We estimate the generalized propensity score using conditional normalizing flows in order to detect regions with limited overlap in the shared covariate-treatment space. (3) We present a gradient-based learning algorithm to find the optimal, individualized dosage combinations. Here, we ensure reliable estimation of the policy value by avoiding regions with limited overlap. We finally perform an extensive evaluation of our method to show its effectiveness. To the best of our knowledge, ours is the first work to provide a method for reliable off-policy learning for optimal dosage combinations.

## 1 Introduction

In personalized medicine, decision-making frequently involves complex choices for ***dosage combinations***. For example, in cancer therapy, patients with chemotherapy or immunotherapy receive a combination of two or three drugs, and, for each, the dosage must be carefully personalized [16, 82]. In critical care, medical professionals must simultaneously control multiple parameters for mechanical ventilation, including respiratory rate and tidal volume [20].

One way to find optimal dosages (i.e., multiple continuous treatments) is through randomized controlled trials. Yet, such randomized controlled trials are often impractical due to the complexity of dosing problems or even unethical [62]. Instead, there is a direct need for learning optimal, individualized dosage combinations from observational data (so-called off-policy learning).

Several methods for off-policy learning aim at discrete treatments [1, 3, 4, 14, 23, 37, 40, 74, 76] and are thus *not* applicable to dosage combinations. In contrast to that, there are only a few methods that aim at off-policy learning for continuous treatments, that is, dosages [6, 10, 41, 44, 87]. To achieve this, some of these methods utilize the individualized dose-response function, which is typically not available and must be estimated from observational data. However, existing works model the effect of multiple continuous treatments *independently*. Because of that, existing works ignore the dependence structure among dosage combinations, that is, drug-drug interactions. For example, in chemotherapy, the combination of antineoplastic agents with drugs such as warfarin can have adverse effects on patient health [61]. In contrast to that, methods for off-policy learning that account for the *joint* effect of dosage combinations have received little attention and present the focus of our paper.

37th Conference on Neural Information Processing Systems (NeurIPS 2023).

Crucially, off-policy learning for dosage combinations comes with non-trivial challenges. (1) Empirically, datasets in medical practice are often high-dimensional, yet of limited sample size. Hence, in practice, there are often sparse regions in the covariate-treatment space, which, in turn, may lead to estimated individualized dose-response functions that have regions with high uncertainty. (2) Mathematically, off-policy learning commonly relies upon the overlap assumption [65]. Yet, certain drug combinations are rarely used in medical practice and should be avoided for safety reasons. Both challenges lead to areas with limited observed overlap in the data, which results in unreliable estimates of the policy value. Motivated by this, we thus develop a method for *reliable* off-policy learning.

**Proposed method:** In this paper, we propose a novel method for reliable off-policy learning for dosage combinations. Our method proceeds along three steps. (1) We develop a tailored neural network that estimates the individualized dose-response function while accounting for the joint effect of multiple dependent dosages. Later, we use the network as a plug-in estimator for the policy value. (2) We estimate the generalized propensity score through conditional normalizing flows in order to detect regions with limited overlap in the shared covariate-treatment space. (3) We find the optimal individualized dosage combinations. To achieve that, we present a gradient-descent-ascent-based learning algorithm that solves the underlying constrained optimization problem, where we avoid regions with limited overlap and thus ensure a reliable estimation of the policy value. For the latter step, we use an additional policy neural network to predict the optimal policy for reasons of scalability.

**Contributions:**[1] (1) We propose a novel method for reliable off-policy learning for dosage combinations. To the best of our knowledge, ours is the first method that accounts for the challenges inherent to dosage combinations. (2) As part of our method, we develop a novel neural network for modeling the individualized dose-response function while accounting for the joint effect of dosage combinations. Thereby, we extend previous works which are limited to mutually exclusive or independent treatments. (3) We perform extensive experiments based on real-world medical data to show the effectiveness of our method.

## 2 Related Work

**Estimating the individualized dose-response function:** There are different methods to estimate the outcomes of continuous treatments, that, is the individualized dose-response function. One stream uses the generalized propensity score [26, 28, 29], while others adapt generative adversarial networks [5]. An even other stream builds upon the idea of shared representation learning from binary treatments (e.g., [33, 53, 68, 69, 84]) and extends it to continuous treatments [57, 67]. However, all of the previous methods model the marginal effects of multiple dosages *independently* and thus *without* considering the dependence among when multiple treatments are applied simultaneously. Yet, this is unrealistic in medical practice as there are generally drug-drug interactions. To address this, we later model the *joint* effect of multiple *dependent* dosages.

There are some approaches for estimating the individualized effects of multi- or high-dimensional treatments, yet *not* for dosages. For example, some methods target conditional multi-cause treatment effect estimation [55, 58, 59, 73, 78, 86] but are restricted to discrete treatments and can *not* be directly applied to dosages. Other works use techniques tailored for high-dimensional structured treatments [21, 36, 54]. However, they do not focus on smooth and continuous estimates, or reliable off-policy learning, which is unlike our method.

**Off-policy learning for dosages:** There is extensive work on off-policy learning for *discrete* treatments [1, 3, 4, 14, 17, 23, 37, 40, 49, 50, 72, 74, 85]. While, in principle, these methods could be applied to our setting by discretizing the dosages, crucial information about the exact individually prescribed dosage values would be lost. This results in less accurate and less personalized dosage recommendations. However, there are only a few papers that address off-policy learning directly for *continuous* treatments ("dosages").

For dosages, previous works have derived efficient estimators for policy evaluation and subsequent optimization (e.g., [10, 41]). Other works focus directly on the policy optimization objective to find an optimal dosage [6, 44, 87]. In principle, some of these methods are applicable to settings with multiple dosages where they act as naïve baselines. However, all of them have crucial shortcomings

---

[1]Code is available at `https://github.com/JSchweisthal/ReliableDosageCombi`.

in that they do *not* account for multiple dependent dosages and/or that they do *not* aim at reliable decision-making under limited overlap. Both of the latter are contributions of our method.

**Dealing with limited overlap:** Various methods have been proposed to address limited overlap for estimating average treatment effects of binary treatments by making use of the propensity score (e.g., [7, 9, 24, 30, 38, 66]). In contrast, there exist only a few works for off-policy learning with limited overlap, typically for binary treatments [39, 42, 70].

Some works deal with the violation of the overlap assumption by leveraging uncertainty estimates for binary [31] and continuous treatment effect estimation [32]. However, these works do not tackle the problem of direct policy learning. In particular for dosage combinations, this is a clear shortcoming, as determining and evaluating policies becomes computationally expensive and infeasible with a growing number of samples and treatment dimensions.

**Research gap:** Personalized decision-making for optimal dosage combinations must (1) estimate the *joint* effect of dosage combinations, (2) then perform off-policy learning for such dosage combinations, and (3) be *reliable* even for limited overlap. However, to the best of our knowledge, there is no existing method for this composite task, and we thus present the first method aimed at reliable off-policy learning for dosage combinations.

## 3 Problem Formulation

**Setting:** We consider an observational dataset for $n$ patients with i.i.d. observations $\{(x_i, t_i, y_i)\}_{i=1}^n$ sampled from a population $(X, T, Y)$, where $X \in \mathcal{X} \subseteq \mathbb{R}^d$ are the patients' covariates, $T \in \mathcal{T} \subseteq \mathbb{R}^p$ is the assigned $p$-dimensional dosage combination (i.e., multiple continuous treatments), and $Y \in \mathbb{R}$ is the observed outcome. For example, in a mechanical ventilation setting, $X$ are patient features and measured biosignals, such as age, sex, respiratory measurements, and cardiac measurements; $T$ are the dosages of the ventilation parameters, such as respiratory rate and tidal volume; and $Y$ is patient survival. For simplicity, we also refer to the dosages as treatments.

We adopt the potential outcomes framework [64], where $Y(t)$ is the potential outcome under treatment $T = t$. In medicine, one is interested in learning a policy $\pi : \mathcal{X} \to \mathcal{T}$ that maps covariates to suggested dosage combinations. Then, the policy value $V(\pi) = \mathbb{E}[Y(\pi(X))]$ is the expected outcome of policy $\pi$. Our objective is to find the optimal policy $\pi^*$ in the policy class $\Pi$ that maximizes $V(\pi)$, i.e.,

$$\pi^* \in \arg\max_{\pi \in \Pi} V(\pi). \tag{1}$$

In other words, we want to find $\pi^*$ that minimizes the regret given by $R(\pi^*) = \max_{\pi \in \Omega} V(\pi) - V(\pi^*)$, where $\Omega$ denotes the set of all possible policies with $\Pi \subseteq \Omega$.

We further introduce the following notation, which is relevant to our method later. We denote the *conditional potential outcome function* by $\mu(t, x) = \mathbb{E}[Y(t) \mid X = x]$. For continuous treatments, this is also called *individualized dose-response function*. Further, the *generalized propensity score* (GPS) is given by $f(t, x) = f_{T \mid X=x}(t)$, where $f_{T \mid X=x}$ denotes the conditional density of $T$ given $X = x$ with respect to the Lebesgue measure. The GPS is the stochastic policy used for assigning the observed treatments (i.e., logging / behavioral policy). In our method, both nuisance functions, $\mu(t, x)$ and $f(t, x)$, are not known and must be later estimated from observational data.

**Assumptions:** To ensure identifiability of the outcome estimation, the following three standard assumptions are commonly made in causal inference [65]: (1) *Consistency*: $Y = Y(T)$. Consistency requires that the patient's observed outcome is equal to the potential outcome under the observed treatment. (2) *Ignorability*: $Y(t) \perp\!\!\!\perp T \mid X, \forall t \in \mathcal{T}$. Ignorability ensures that there are no unobserved confounders. (3) *Overlap*: $f(t, x) > \varepsilon, \forall x \in \mathcal{X}, t \in \mathcal{T}$ and for some constant $\varepsilon \in [0, \infty)$. Overlap (also known as common support or positivity) is necessary to ensure that the outcomes for all potential treatments can be estimated accurately for all individuals.

We distinguish between *weak* overlap for $\varepsilon = 0$ and *strong* overlap for $\varepsilon > 0$. Since the estimation of the individualized dose-response function becomes more reliable with increasing $\varepsilon$ in the finite sample setting, we refer to *strong* overlap in the following when we mention overlap, unless explicitly stated otherwise.

**Challenges due to limited overlap:** Limited overlap introduces both empirical and theoretical challenges. (1) Datasets with high dimensions and/or small sample size are especially prone to have

sparse regions in the covariate-treatment space $\mathcal{X} \times \mathcal{T}$ and thus limited overlap [9]. As a result, the estimated individualized dose-response function will have regions with high uncertainty, which preclude reliable decision-making. (2) It is likely that $\varepsilon$ is small for some regions as some drug combinations are rarely prescribed and should be avoided. To address this, we aim at an off-policy method that is reliable, as introduced in Sec. 4.

## 4 Reliable Off-Policy Learning for Dosage Combinations

In the following, we propose a novel method for reliable off-policy learning for dosage combinations. Our method proceeds along three steps (see Fig. 1): **(1)** We develop a tailored neural network that estimates the individualized dose-response function while accounting for the joint effect of multiple dependent dosages (Sec. 4.1). Later, we use the network as a plug-in estimator for the policy value. **(2)** We estimate the generalized propensity score through conditional normalizing flows in order to detect regions with limited overlap in the shared covariate-treatment space (Sec. 4.2). **(3)** We find the optimal individualized dosage combinations (Sec. 4.3). To achieve that, we present a gradient-descent-ascent-based learning algorithm that solves the underlying constrained optimization problem (Sec. 4.4), where we avoid regions with limited overlap and thus ensure a reliable estimation of the policy value. For the latter step, we use an additional policy neural network to predict the optimal policy for fast inference during deployment.

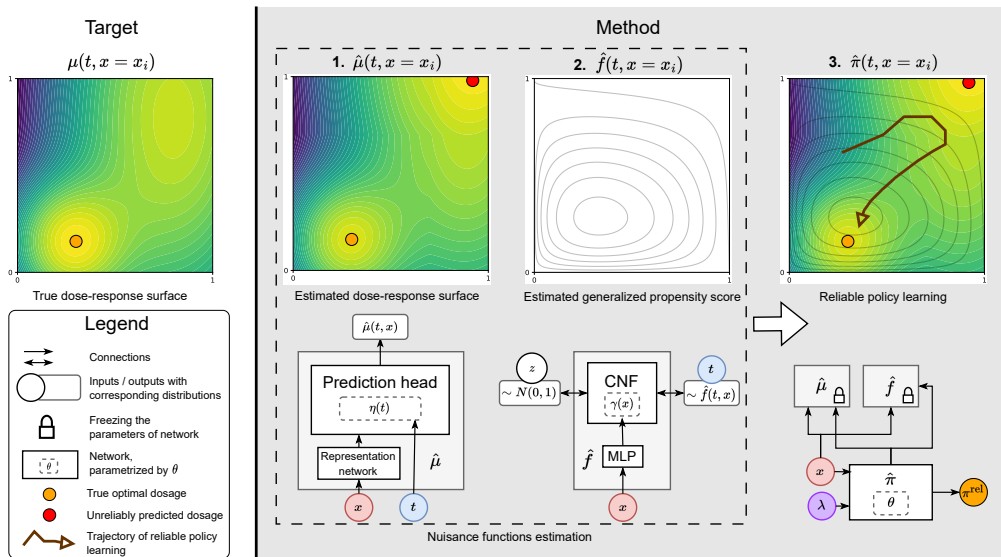

Figure 1: Overview of our method along steps (1)–(3).

### 4.1 Neural network for estimating the joint effect of multiple dependent dosages

In the first step, we aim to estimate the individualized dose-response function $\mu(t, x)$, which is later used as a plug-in estimator for the policy value $V(\pi)$. However, using a naïve neural network with $t$ and $x$ concatenated as inputs can lead to diminishing effect estimates of $t$ when $x$ is high-dimensional [68]. In addition, in our setting with multiple continuous treatments, we want to yield smooth estimates of the dose-response surface to ensure better generalization when interpolating between sparse observations. To address this, we thus develop a tailored *dosage combination network* (DCNet) in the following.

**Architecture of DCNet:** Our DCNet builds upon the varying coefficient network (VCNet) [56] but extends it to learn the *joint* effect of *multiple* dependent dosages. DCNet consists of (i) a representation network $\phi(x)$ and (ii) a prediction head $h_{\eta(t)}(\phi)$, where $h$ is a network with $d_\eta$ different parameters $\eta(t) = \left(\eta_1(t), \ldots, \eta_{d_\eta}(t)\right)^T$. The parameters $\eta(t)$ are not fixed but depend on the dosage $t$. This is crucial to ensure that the influence of $t$ does not diminish in a setting with high-dimensional covariates $x$.

To handle multiple dosages $(t^{(1)}, \ldots, t^{(p)})$, a naïve way [2] would be to incorporate a separate prediction head for each treatment with corresponding parameters $\eta^{(i)}(t^{(i)})$. However, this would be ineffective for our task, as it captures only the marginal effect of a dosage but not the joint effect of the dosage combination. Instead, in order to learn the joint effect, we must not only model the marginal effect of a dosage (as in [2]) but also drug-drug interactions. We achieve that through a custom prediction head.

**Prediction head in DCNet:** In DCNet, we use a tailored prediction head, which is designed to model the *joint* effect of dosage combinations. For this, we leverage a tensor product basis (see, e.g., [48, 81]) to estimate smooth interaction effects in the dose-response surface. As a result, we can directly incorporate all $p$ dosages simultaneously into the parametrization of the network $h$ by using only one head containing the joint dosage information. This is unlike other networks that use $p$ different independent heads [2] and thus cannot capture the joint effect but only marginal effects.

We define each scalar parameter $\eta_j(t)$ of our prediction head as

$$
\eta_j(t) = \sum_{k_1=1}^{K_1} \sum_{k_2=1}^{K_2} \cdots \sum_{k_p=1}^{K_p} \beta_{j,k_1 k_2 \ldots k_p} \cdot \psi_{k_1}^{(1)}(t^{(1)}) \cdot \psi_{k_2}^{(2)}(t^{(2)}) \cdot \ldots \cdot \psi_{k_p}^{(p)}(t^{(p)}), \tag{2}
$$

where $K_1, \ldots, K_p$ are the number of basis functions $\psi_{k_i}^{(i)}$ of the $p$ dosages $t^{(1)}, \ldots, t^{(p)}$, and $\beta_{j,\ldots}$ are the trainable coefficients for each element of the tensor product. In order to preserve the continuity of the non-linear dose-response surface, we choose $\psi_{k_i}^{(i)}$ to be polynomial spline basis functions. Hence, we yield the parameter vector $\eta(t) = \mathbf{B}\boldsymbol{\Psi}(t)$, where

$$
\mathbf{B} = \begin{bmatrix} \beta_{1,1\ldots1} & \cdots & \beta_{1,K_1\ldots K_p} \\ \vdots & \ddots & \vdots \\ \beta_{d_\eta,1\ldots1} & \cdots & \beta_{d_\eta,K_1\ldots K_p} \end{bmatrix} \in \mathbb{R}^{d_\eta \times (\prod_{i=1}^p K_i)} \tag{3}
$$

with matrix rows arranged as $(\beta_{j,11\ldots1}, \ldots, \beta_{j,K_1 1\ldots1}, \ldots, \beta_{j,K_1\ldots K_{p-1}1}, \ldots, \beta_{j,K_1\ldots K_{p-1}K_p})$, with

$$
\boldsymbol{\Psi}(t) = \left( \psi^{(1)}(t^{(1)}) \otimes \psi^{(2)}(t^{(2)}) \otimes \cdots \otimes \psi^{(p)}(t^{(p)}) \right) \in \mathbb{R}^{(\prod_{i=1}^p K_i)} \tag{4}
$$

and with $\psi^{(i)} = (\psi_1^{(i)}, \ldots, \psi_{K_i}^{(i)})^T$, where $\otimes$ is the Kronecker product. For training DCNet, we minimize the mean squared error loss $\mathcal{L}_\mu = \frac{1}{n} \sum_{i=1}^n (\hat{\mu}(t_i, x_i) - y_i)^2$.

Our prediction head has three key benefits: (1) When estimating $\mu(t, x)$, the tensor product basis enables us to model non-linear interaction effects among different dosage dimensions while still having a dose-response surface that offers continuity, smoothness, and expressiveness. (2) Having a dose-response surface that is smooth between different dosages is beneficial for our task as it allows for better interpolation between sparse data points. (3) Smoothness is also exploited by our gradient-based learning algorithm later. The reason is that a smooth plug-in estimator $\hat{\mu}(t, x)$ for the estimated policy value $\hat{V}(\pi)$ allows us to have more stable gradients and thus more stable parameter updates during learning.

## 4.2 Conditional normalizing flows for detecting limited overlap

In the second step, we estimate the GPS $f(t, x)$ in order to detect limited overlap, which later allows us to avoid regions with high uncertainty and thus to yield a reliable policy.

*Why is the GPS relevant for reliable decision-making?* Due to sparsity in the data or otherwise limited overlap, the estimates of the individualized dose-response function $\hat{\mu}(t, x)$ from our DCNet may be naturally characterized by regions with high uncertainty, that is, regions in the covariate-treatment space with low GPS. If we would use the estimated $\hat{\mu}(t, x)$ in a naïve manner as a plug-in estimator for policy learning, we may yield a policy where there can be a large discrepancy between the predicted outcome $\hat{\mu}(t, x)$ and the true outcome $\mu(t, x)$ and, therefore, a large discrepancy between $\hat{V}(\pi)$ and $V(\pi)$. By avoiding regions with a low GPS, we can thus ensure reliable decision-making in safety-critical applications such as medicine.

**Estimation of GPS:** The GPS (i.e., the logging policy) is typically not known in real-world settings. Instead, we must estimate the GPS from observational data. To this end, we leverage *conditional*

*normalizing flows* (CNFs) [75, 80]. CNFs are a fully-parametric generative model built on top of normalizing flows [60, 71] that can model conditional densities $p(y \mid x)$ by transforming a simple base density $p(z)$ through an invertible transformation with parameters $\gamma(x)$ that depend on the input $x$.[2]

In our method, we use CNFs with parameters $\gamma(x)$ to estimate the probability density function $f(t, x) = f_{T \mid X=x}(t)$ of $T$ conditioned on the patient covariates $X = x$. For our method, CNFs have several advantages.[3] First, CNFs are universal density approximators [11, 12, 15, 27], making them flexible and able to capture even complex density functions for dosage combinations. Second, CNFs are properly normalized. Third, unlike kernel methods, CNFs are fully parametric, and, once trained, the inference time is constant [52]. This is particularly important later in the third step of our method (Sec. 4.3): For each training step and training sample in the batch, the predicted conditional density has to be evaluated again under the updated policy, and, hence, a fast inference time of density is crucial to achieving a significant speed up when training the policy network.

In our method, we use *neural spline flows* [15], as they allow for flexible estimation of multimodal densities, and, in combination with masked auto-regressive networks [13], we can extend them to multiple dimensions. This combination allows for fast and exact evaluation of the multidimensional densities. We also set the base density, $p(z)$, to standard normal distribution. The CNFs are trained by minimizing the negative log-likelihood loss $\mathcal{L}_f = -\frac{1}{n} \sum_{i=1}^{n} \log \hat{f}(t_i, x_i)$ .

In the following section, we use the estimated GPS $\hat{f}(t, x)$ during policy learning in order to avoid regions with limited overlap and thus ensure reliable decision-making.

### 4.3 Constrained optimization to avoid regions with limited overlap

**Reliable decision-making**: Our aim is to find the policy $\pi^{\text{rel}}$ which maximizes the estimated policy value $\hat{V}(\pi)$ while ensuring that $V(\pi)$ can be estimated *reliably*. For this, we suggest to constrain our search space in the shared covariate-treatment space to regions that are sufficiently supported by data, i.e., where overlap is not violated. Formally, we rewrite our objective from Eq. (1) as

$$\pi^{\text{rel}} \in \arg\max_{\pi \in \Pi^{\text{r}}} \hat{V}(\pi) \qquad \text{with} \quad \Pi^{\text{r}} = \{\pi \in \Pi \mid f(\pi(x), x) > \bar{\varepsilon}), \ \forall x \in \mathcal{X}\}, \qquad (5)$$

where $\bar{\varepsilon}$ is a reliability threshold controlling the minimum overlap required for our policy. In our setting with finite observational data, this yields the constrained optimization problem

$$\max_{\pi} \quad \frac{1}{n} \sum_{i=1}^{n} \hat{\mu}(\pi(x_i), x_i) \quad \text{s.t.} \quad \hat{f}(\pi(x_i), x_i) \geq \bar{\varepsilon}, \ \forall i \in \{1, \ldots, n\}. \qquad (6)$$

Here, we use our DCNet as a plug-in dose-response estimator $\hat{\mu}(t, x)$, and we further use our conditional normalizing flows to restrict the search space to policies with a sufficiently large estimated GPS $\hat{f}(t, x)$.

**Neural network for reliable policy learning:** To learn the optimal reliable policy, we further suggest an efficient procedure based on gradient updates in the following. Specifically, we train an additional policy neural network to predict the optimal policy. Our choice has direct benefits in practice. *Why not just optimize over the dosage combinations for each patient individually?* In principle, one could also use methods for constrained optimization and leverage the nuisance functions $\hat{\mu}$ and $\hat{f}$ to directly optimize over the dosage assignments $t$ for each patient, i.e., $t_i^{\text{rel}} = \arg\max_{\tilde{t}_i} \left[ \hat{\mu}(\tilde{t}_i, x_i) \right]$ s.t. $\hat{f}(\tilde{t}_i, x_i) \geq \bar{\varepsilon}$. However, this does *not* warrant scalability at inference time, as it requires that one solves the optimization problem for *each* incoming patient. The complexity of this optimization problem scales with both the number of dosages and the number of samples in the data to be evaluated. This makes treatment recommendations costly, which prohibits real-time

---

[2]Normalizing flows were introduced for expressive variational approximations in variational autoencoders [60, 71]. We provide a background in Appendix A.

[3]Most prior works neglect a proper estimation of the GPS by mainly using non-flexible parametric models [26] or kernel density estimators [41]. Similarly, the varying coefficient network from [56] uses an additional density head but enforces a discretization. It is thus overly simplified and, therefore, *not* reliable and *not* continuous. Both are crucial shortcomings for our task. As a remedy, we propose the use of CNFs.

decision-making in critical care. As such, direct methods for optimization are rendered impractical for real-world medical applications. Instead, we suggest to directly learn a policy that predicts the optimal dosage combination for incoming patients.

As the constrained optimization problem in Eq. (6) cannot be natively learned by gradient updates, we must first transform Eq. (6) into an unconstrained Lagrangian problem

$$\min_\theta \; -\frac{1}{n}\sum_{i=1}^n \hat{\mu}\left(\pi_\theta(x_i), x_i\right) \quad \text{s.t. } \hat{f}\left(\pi_\theta(x_i), x_i\right) \geq \overline{\varepsilon}, \forall i$$

$$\iff \min_\theta \max_{\lambda_i \geq 0} -\frac{1}{n}\sum_{i=1}^n \left\{ \hat{\mu}\left(\pi_\theta(x_i), x_i\right) - \lambda_i \left[ \hat{f}\left(\pi_\theta(x_i), x_i\right) - \overline{\varepsilon} \right] \right\}, \tag{7}$$

where $\pi_\theta(x_i)$ is the policy learner with parameters $\theta$, and $\lambda_i$ are the Lagrange multipliers for sample $i$. The Lagrangian min-max-objective can be solved by adversarial learning using gradient descent-ascent optimization (see, e.g., [47] for a background). Details are presented in the next section where we introduce our learning algorithm.

### 4.4 Learning algorithm

The learning algorithm for our method proceeds along the following steps (see Algorithm 1):

• (1) We use our DCNet to estimate the individualized dose-response function $\hat{\mu}(t_i, x_i)$ for dosage combinations.

• (2) We estimate the GPS $\hat{f}(t_i, x_i)$ using the CNFs to detect areas with limited overlap.

• (3) We plug the estimated $\hat{\mu}(t, x)$ and $\hat{f}(t, x)$ into the Lagrangian objective from Eq. (7). As the policy learner, we use a multilayer-perceptron neural network $\hat{\pi}_\theta(x)$ to predict the optimal dosage combination $t$ given the patient covariates $x$. We use stochastic gradient descent-ascent. In detail, we perform adversarial learning with alternating parameter updates. First, in every training step, we perform a gradient *descent* step to update the parameters $\theta$ of the policy network $\hat{\pi}_\theta$ with respect to the policy loss

$$\mathcal{L}_\pi(\theta, \lambda) = -\frac{1}{n}\sum_{i=1}^n \left\{ \hat{\mu}\left(\hat{\pi}_\theta(x_i), x_i\right) - \lambda_i \left[ \hat{f}\left(\hat{\pi}_\theta(x_i), x_i\right) - \overline{\varepsilon} \right] \right\}, \tag{8}$$

where $\overline{\varepsilon}$ is the reliability threshold and where $\lambda = \{\lambda_i\}_{i=1}^n$ are trainable penalty parameters. Then, we perform a gradient *ascent* step to update the penalty parameters $\lambda$ with respect to $\mathcal{L}_\pi$. Here, each $\lambda_i$ is optimized for each sample $i$ in the training set.

As there is no guarantee for global convergence in the non-convex optimization problem, we perform $k = 1, \ldots, K$ random restarts and select the best run as evaluated on the validation set by

$$\pi_\theta^{\text{rel}} = \pi_\theta^{(j)}, \quad \text{with} \quad j = \arg\max_k \sum_{i=1}^n \hat{\mu}\left(\pi_\theta^{(k)}(x_i), x_i\right) \cdot \mathbb{1}\left\{ \hat{f}\left(\pi_\theta^{(k)}(x_i), x_i\right) \geq \overline{\varepsilon} \right\}, \tag{9}$$

where $\pi_\theta^{(k)}$ is the learned policy in run $k$ and $\mathbb{1}\{\cdot\}$ denotes the indicator function. As a result, we select the policy $\pi_\theta^{(j)}$ which maximizes $\hat{V}(\pi)$ under the overlap constraint $\hat{f}\left(\pi_\theta^{(k)}(x_i), x_i\right) \geq \overline{\varepsilon}, \; \forall i$. Note that we use the validation loss from Eq. (9) because $\mathcal{L}_\pi$ cannot be directly applied for evaluating the performance on the validation set, as $\lambda_i$ can only be optimized for observations in the training set during the policy learning.

## 5 Experiments

We perform extensive experiments using semi-synthetic data from real-world medical settings to evaluate the effectiveness of our method. Semi-synthetic data are commonly used to evaluate causal inference methods as it ensures that the causal ground truth is available [8, 83] and thus allows us to benchmark the performance.

**MIMIC-IV:** MIMIC-IV [35] is a state-of-the-art dataset with de-identified health records from patients admitted to intensive care units. Analogous to [67], we aim to learn optimal configurations of mechanical ventilation to maximize patient survival ($Y$). Here, treatments ($T$) involve the configuration of mechanical ventilation in terms of respiratory rate and tidal volume. When filtering for $T$, we yield a dataset with $n = 5476$ patients. We select 33 variables as patient covariates $X$ (e.g., age, sex, respiratory measurements, cardiac measurements). Unlike [67] where treatments $T$ are modeled as $p$ mutually exclusive dosages, we model the more realistic setting, where $T$ are dosage combinations with a joint effect. Our choice is well aligned with medical literature according to which the parameters of mechanical ventilation are dependent and must be simultaneously controlled [20]. Further details for MIMIC-IV are in Appendix B.

**TCGA:** TCGA [79] is a diverse collection of gene expression data from patients with different cancer types. Our aim is to learn the optimal dosage combination for chemotherapy ($T$). In chemotherapy, combining different drugs is the leading clinical option for treating cancer [82]. Let patient survival denote the outcome ($Y$). We select the same version of the dataset as in [5, 67], containing the measurement of 4,000 genes as features $X$ from $n = 9659$ patients. Unlike previous works [5, 67], we again model $T$ as different, simultaneously assigned dosages with a joint effect. This follows medical practice where drug combinations in cancer therapy must be jointly optimized for each patient profile [16, 82]. Given that TCGA is high-dimensional, we expect it to be an especially challenging setting for evaluation. Further details for TCGA are in Appendix B.

In both datasets, the response $Y \sim N(\mu(T, X), 0.5)$ can be interpreted as the log-odds ratio of patient survival, which is to be maximized. Due to the fact that we work with actual data from medical prac-

---

**Algorithm 1:** Reliable off-policy learning for dosage combinations

**Input** : data $(X, T, Y)$, reliability threshold $\overline{\varepsilon}$
**Output:** optimal reliable policy $\hat{\pi}_\theta^{\mathrm{rel}}$
// Step 1: Estimate individualized dose-response function using our DCNet
Estimate $\hat{\mu}(t, x)$ via loss $\mathcal{L}_\mu$
// Step 2: Estimate GPS using conditional normalizing flows
Estimate $\hat{f}(t, x)$ via loss $\mathcal{L}_f$
// Step 3: Train policy network using our reliable learning algorithm
**for** $k \in \{1, 2, \ldots, K\}$ **do**
  // K runs with random initialization
  $\hat{\pi}_\theta^{(k)} \leftarrow$ initialize randomly
  $\lambda \leftarrow$ initialize randomly
  **for** *each epoch* **do**
    **for** *each batch* **do**
      $\mathcal{L}_\pi \leftarrow -\frac{1}{n} \sum_{i=1}^n \left\{ \hat{\mu}\left(\hat{\pi}_\theta^{(k)}(x_i), x_i\right) \right.$
      $\left. -\lambda_i \left[ \hat{f}\left(\hat{\pi}_\theta^{(k)}(x_i), x_i\right) - \overline{\varepsilon} \right] \right\}$
      $\theta \leftarrow \theta - \eta_\theta \nabla_\theta \mathcal{L}_\pi$
      $\lambda \leftarrow \lambda + \eta_\lambda \nabla_\lambda \mathcal{L}_\pi$
    **end**
  **end**
**end**
// select best learned policy wrt constrained objective on validation set
$\hat{\pi}_\theta^{\mathrm{rel}} \leftarrow \pi_\theta^{(j)}$, with $j =$
$\arg\max_k \sum_{i=1}^n \hat{\mu}\left(\pi_\theta^{(k)}(x_i), x_i\right) \cdot \mathbb{1}\left\{ \hat{f}\left(\pi_\theta^{(k)}(x_i), x_i\right) \geq \overline{\varepsilon} \right\}$

---

tice, certain dosage combinations conditional on the patients' features are more likely to be observed. Even other dosage combinations may not be assigned at all as they can cause harm. For our experiments, we introduce a parameter $\alpha$ controlling the dosage bias, which changes the observed policy of dosage assignments from random uniform ($\alpha = 0$) to completely deterministic ($\alpha = \infty$). Hence, with increasing $\alpha$, the areas in the covariate-treatment space will be increasingly prone to limited overlap. We refer to Appendix B for details. When not stated otherwise, we choose $\alpha = 2$ and $p = 2$ as the default setting for our experiments.

**Baselines:** To the best of our knowledge, there are no existing methods tailored to off-policy learning for dosage combinations under limited overlap. Hence, we compare our method with the following baselines, which vary different steps in our method and thus allow us to attribute the source of performance gain. Specifically, we vary dose-response estimation ($\hat{\mu}(t, x)$) vs. optimization ($\hat{\pi}$). (1) For estimating $\hat{\mu}(t, x)$, we use two baselines: (i) a simple multi-layer perceptron (**MLP**) as a standard neural alternative to our DCNet, and (ii) **VCNet** [56] as the SOTA method for dose-response estimation for *independent* treatments with a separate head per treatment dimension. (2) For optimization, we minimize the policy loss $-\frac{1}{n} \sum_{i=1}^n \hat{\mu}\left(\hat{\pi}_\theta(x_i), x_i\right)$ without constraints for dealing with limited overlap (**naïve**).

**Evaluation:** For training, we split the data into train / val / test sets (64 / 16 / 20%). We first train the nuisance models $\hat{\mu}(t, x)$ and $\hat{f}(t, x)$. We then perform $k = 5$ random restarts to train $\hat{\pi}(t, x)$. We select the best run with respect to the policy loss on the factual validation dataset. Thereby, we carefully adhere to the characteristics in real-world settings, where one does *not* have access to the true dose-response function $\mu(t, x)$ on the validation set for model selection. In our method, we set

the reliability threshold $\bar{\varepsilon}$ to the 5%-quantile of the estimated GPS $\hat{f}(t, x)$ of the train set. Further details including hyperparameter tuning are in Appendix E.

For evaluation, we compare the methods on the test set using the regret $R(\hat{\pi}) = \max_{\pi \in \Omega} V(\pi) - V(\hat{\pi})$. We report: (1) the regret for the selected policy, which is the policy with the best policy loss on the validation set out of all $k$ policies; (2) the average regret across all $k$ restarts; and (3) the standard deviation to better assess the reliability of the method.

**Results:** Table 1 compares the performance of our method against the baselines. We make the following observations: (1) DC-Net performs similarly to MLP for low-dimensional data such as MIMIC-IV, as expected. However, we find substantial improvements from DCNet over MLP for high-dimensional data such as TCGA, showing the advantage of enforcing smoothness and expressiveness in our prediction head. (2) DCNet outperforms VCNet on both

| Methods | | MIMIC-IV | | | TCGA | | |
|---------|---|----------|------|-----|------|------|-----|
| $\hat{\mu}$ | $\hat{\pi}$ | Selected | Mean | Std | Selected | Mean | Std |
| Oracle ($\mu$) | observed | 1.21 | – | – | 1.06 | – | – |
| MLP | naïve | 0.02 | 1.44 | 1.56 | 1.94 | 1.88 | 0.12 |
| MLP | reliable | 0.03 | 0.03 | 0.00 | 1.78 | 1.78 | 0.03 |
| VCNet | naïve | 2.13 | 2.49 | 0.92 | 3.64 | 2.47 | 1.19 |
| VCNet | reliable | 0.07 | 0.07 | 0.00 | 0.06 | 0.06 | 0.00 |
| DCNet | naïve | 2.81 | 2.07 | 1.17 | 2.52 | 0.94 | 1.21 |
| DCNet | reliable (**ours**) | 0.04 | 0.04 | 0.00 | 0.03 | 0.03 | 0.00 |

Reported: best / mean / standard deviation (lower = better)

Table 1: Performance against baselines. Regret on test set over $k = 5$ restarts.

datasets, which demonstrates the importance of modeling the joint effect of dosage combinations. (3) Choosing a policy that is reliable greatly benefits overall performance. For example, the baselines (+naïve) even perform worse than the observed policy for TCGA, whereas our method (+reliable) is clearly superior. (4) We also observe that our selection of reliable policies greatly reduces the standard deviation of the observed regrets. This further demonstrates the reliability of our proposed method. Overall, our method works best across both datasets. For example, the regret on the TCGA dataset for the selected policy drops from 2.52 (DCNet+naïve) to only 0.03 (DCNet+reliable), which is a reduction by 98.8%. In sum, this demonstrates the effectiveness of our method.

**Comparison to discretization:** There exist several methods for off-policy learning for discrete treatments (see Sec. 2), including methods targeted for overlap violations and treatment combinations. In principle, we could also discretize the dosages in our setting and apply such methods. However, we would then lose information about the exact dosage values and could not exploit the continuity of the dose-response surface. To demonstrate the benefits of our method, we benchmark against an "oracle" discretized baseline where we assume perfect knowledge of the dose-response function and perfect individual treatment assignment. This serves as an upper bound for all policy learning methods with discretization. We display the results for different granularities of equally-spaced grid discretization in Table 2. We observe that, even in the oracle setting, discretization itself leads to a higher regret than our proposed method, which confirms the advantages of leveraging continuous dosage information.

| Methods | | MIMIC-IV | TCGA |
|---------|---|----------|------|
| $\hat{\mu}$ | $\hat{\pi}$ | Regret | |
| Oracle ($\mu$) | discrete (3x3) | 3.32 | 2.94 |
| Oracle ($\mu$) | discrete (4x4) | 0.96 | 0.95 |
| Oracle ($\mu$) | discrete (5x5) | 0.05 | 0.06 |
| DCNet | reliable (**ours**) | 0.04 | 0.03 |

Reported: best / mean / standard deviation (lower = better)

Table 2: Performance against methods based on discretized treatments.

**Robustness checks:** We evaluate the robustness of our method across settings with different levels of limited overlap. For this, we vary the dosage bias $\alpha$ (in Fig. 2) and the number of dosages $p$ (in Fig. 3). Here, we use DCNet for modeling $\hat{\mu}(t, x)$ in both methods, but then vary the policy selection (naïve vs. reliable, as defined above). Our method (DCNet+reliable) shows robust behavior. As desired, it achieves a low regret and a low variation in all settings. In contrast, the naïve method leads to high variation. This must be solely attributed to that our constrained optimization for avoiding regions with limited overlap leads to large performance gains. We also observe another disadvantage of the naïve method: it may select a suboptimal policy out of the different runs due to the fact that the estimation of $\hat{\mu}(t, x)$ is erroneous in regions with limited overlap. Again, this underlines the benefits of our method for reliable off-policy learning of dosage combinations in settings with limited overlap. In Appendix C, we demonstrate the robustness of our method in further setups.

Shown: selected policy (line) and the range over 5 runs (area). The naïve baseline has a large variability across runs while ours is highly robust as expected (i.e., we see no variability).

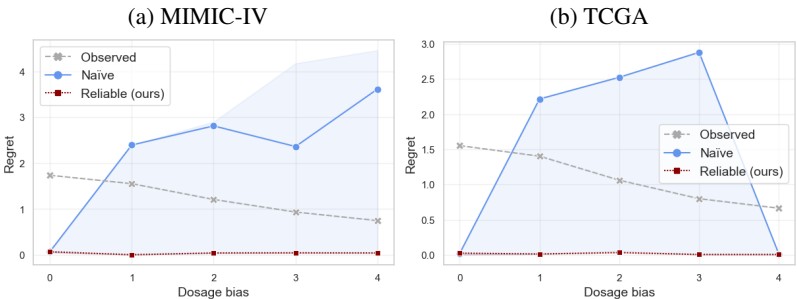

Figure 2: Robustness for dosage bias $\alpha$.

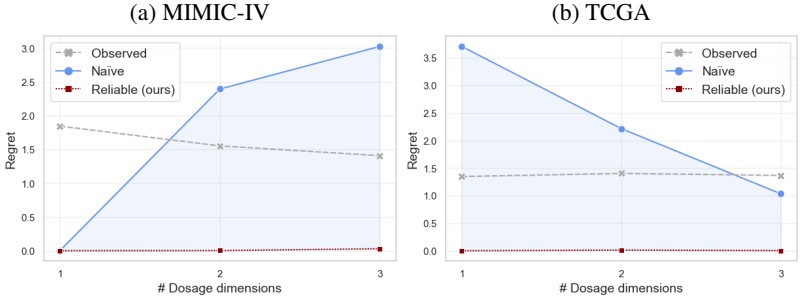

Figure 3: Robustness for number of dosages $p$.

## 6 Discussion

**Limitations:** Our method makes an important contribution over existing literature in personalized treatment design for dosage combinations by adjusting for the naturally occurring but non-trivial problems of drug-drug interactions and limited overlap. However, the complexity of real-world data in medical practice can limit the applicability of our method in certain ways. (i) Our method does not account for the ignorability assumption, which can result in biased estimates in the case of unobserved confounders or other missing data. (ii) We apply our method in a static treatment setting, whereas, in several clinical applications, time-series data are available, e.g., sequences of varying dosages, multiple treatment cycles, and right-censored data. Thus, future work could extend our method to account for causal effect estimates in a time-varying setting (e.g., [18, 25, 51]). (iii) Our method relies on the – for medical applications reasonable – implicit assumption of smooth dose-response surfaces. In settings with extremely unsmooth or even stepwise dose-response surfaces, other methods may be more suitable for dose-response estimation. (iv) Our method aims at reliable policy learning by avoiding areas with limited overlap to minimize potential harm. In other applications such as marketing, one may want to explicitly target unreliable regions to acquire new customers.

**Broader impact:** Our method can have a significant impact in the field of medicine, where reliable dosing recommendations are crucial for effective treatment. Our approach addresses the limitations of previous methods in terms of reliability and estimating joint effects of dosage combinations, and, hence, can improve decision support for clinicians. The reliability of dosage recommendations is particularly important, as incorrect dosing can have serious consequences for patients, such as adverse effects or treatment failure. By addressing this gap, our approach provides a valuable tool for decision support in medicine, empowering healthcare professionals to make more personalized and safe treatment recommendations.

## Acknowledgments

SF acknowledges funding from the Swiss National Science Foundation (SNSF) via Grant 186932.

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

# A  Additional Background

**Estimating the dose-response function under dosage bias:** Estimating the dose-response function becomes challenging with increasing dosage bias. Although the unbiasedness of counterfactual estimation can be achieved through the minimization of the prediction loss on the factual data, the inherent scarcity of data in continuous treatment settings leads to increased variance in the estimation process (see, e.g., [2]), especially in regions with limited overlap. Consequently, the elevated variance hinders accurate generalization over the whole covariate-treatment space and contributes to a larger generalization error, which, in turn, impedes the reliable estimation of treatment effects. Effectively addressing this issue is crucial to improve the robustness and validity of counterfactual predictions.

In Sec. 2, we introduced various existing methods for estimating the dose-response function, e.g., [5, 26, 28, 29, 57, 67]. Besides a custom neural architecture, some methods use additional techniques to reduce the variance of counterfactual predictions for discrete or continuous treatments. Examples include reweighting [63, 77], importance sampling [22], targeted regularization [56], or balancing representations [2, 33, 67]. In our DCNet, we build on top of the neural architectures of [57, 67] and then develop a method that is capable of modeling the joint effect of dosage combinations.

In theory, we could also adapt additional techniques for variance reduction of the dose-response prediction to our setting of dosage combinations and leverage them in combination with DCNet. However, this may not be beneficial for our goal of reliable policy learning for dosage combinations under limited overlap, which is different from the goal of accurate dose-response estimation over the whole covariate-treatment space. These techniques aim to reduce the variance of the dose-response prediction in scarce regions (i.e., with limited overlap), which, as a consequence, shifts the focus away from regions with sufficient overlap. As we constrain the policy to regions with sufficient overlap anyway by using our reliable policy optimization, we aim for an adequate dose-response estimation in these regions, while regions with limited overlap can be neglected. Hence, in our method for reliable policy learning, techniques for tackling dosage bias during dose-response estimation can be considered more counterproductive than useful. We demonstrate this empirically in Appendix D.

**Normalizing flows:** Normalizing flows were introduced for expressive variational approximations in variational autoencoders [60, 71] and are based on the principle of transforming a simple base distribution, such as a Gaussian, through a series of invertible transformations to approximate the target data distribution. Let us denote the base distribution as $p_z(z)$, where $z$ is a latent variable, and the target data distribution as $p_x(x)$, where $x$ represents the observed data. The goal is to find a mapping $f$ such that $x = f(z)$.

The core idea behind normalizing flows is to model the transformation $f$ as a composition of invertible functions $f = f_K \circ f_{K-1} \circ \ldots \circ f_1$, where $K$ is the number of transformations. The last transformation $f_K$ is typically modeled as an affine transformation, and others are invertible nonlinear transformations. The inverse transformation can be easily computed since each $f_k$ is invertible.

The change of variables formula allows us to compute the probability density of $x$ in terms of $z$. If we assume the base distribution $p_z(z)$ is known and has a simple form (e.g., a Gaussian), we can compute the density of $x$ via

$$p_x(x) = p_z(z) \left| \det \left( \frac{\partial f}{\partial z} \right) \right|^{-1} \tag{10}$$

where $\left| \det \left( \frac{\partial f}{\partial z} \right) \right|$ is the determinant of the Jacobian matrix of the transformation $f$ with respect to $z$. This determinant captures the stretching and squishing of the latent space induced by the transformation, thus allowing us to adjust the density of $z$ to match the density of $x$.

To perform inference, such as density estimation or sampling, we need to compute the log-likelihood of the observed data $x$. Given a dataset $\mathcal{D} = \{x_1, \ldots, x_n\}$, the log-likelihood can be computed as the sum of the log-densities of each data point

$$\log p(\mathcal{D}) = \sum_{i=1}^{n} \log p_x(x_i). \tag{11}$$

**Conditional normalizing flows:** In our method, we leverage conditional normalizing flows (CNFs) [75, 80] to estimate the generalized propensity score (GPS). CNFs model conditional densities

$p(y \mid x)$ by transforming a simple base density $p(z)$ through an invertible transformation with parameters $\gamma(x)$ that depend on the input $x$.

In our method, we use neural spline flows [15] in combination with masked auto-regressive networks [13]. Here, masked auto-regressive networks are used to model the conditional distribution of each variable in the data given the conditioning variable. The network takes as input the conditioning variable and generates each variable in the data sequentially, one at a time, while conditioning on the previously generated variables. This autoregressive property allows for efficient computation and generation of samples from the conditional distribution. For the motivation and advantages of our modeling choice, we refer to Sec. 4.2.

**Offline reinforcement learning:** A related literature stream covers off-policy evaluation (OPE) in an offline reinforcement learning (ORL) setting. Some works leverage value functions and/or logging policies to achieve reliable policy learning (e.g., [19, 45, 46]). The main differences to our setting are that ORL assumes a Markov decision process and sequential decision-making. In contrast, we focus on non-sequential, off-policy learning from observational data. Also, unlike ORL or standard OPE, we leverage the causal structure of treatment assignment and the dose-response function to return causal estimands. This allows us not only to learn decisions but also causal estimates for potential outcomes. This is crucial for medical practice, where physicians would like to reason over different treatments rather than blindly following a fully automated system.

## B    Data Description

**MIMIC-IV:** MIMIC-IV[4] is the latest version of the publicly available dataset of The Medical Information Mart for Intensive Care [35] and contains de-identified health records of patients admitted to intensive care units (ICUs). The dataset includes comprehensive clinical data, such as vital signs, laboratory measurements, medications, diagnoses, procedures, and more. In line with the approach in [67], our primary objective is to acquire knowledge about the optimal configurations of mechanical ventilation that can effectively maximize patient survival ($Y$). Note that some works (e.g., [5, 67]) used the previous version of the MIMIC dataset (MIMIC-III [34]), whereas we use the most current one to offer realistic, comparable, and meaningful findings.

As treatments ($T$), we select the specific settings (dosages) utilized for mechanical ventilation, encompassing parameters such as respiratory rate and tidal volume. By filtering the data for the last available measurements per patient before ventilation, we are able to compile a dataset consisting of $n = 5476$ patients. We carefully selected 33 patient covariates ($X$) that encompass factors of medical relevance such as age, sex, and various respiratory and cardiac measurements. For detailed identifiers of the selected covariates, we refer to our code.

While [67] adopted a framework in which treatments $T$ were modeled as a set of $p$ mutually exclusive dosages, we opted for a more realistic approach, considering that the configuration of mechanical ventilation involves dosage combinations with a joint effect. This decision aligns closely with existing medical literature, which emphasizes the interdependence of mechanical ventilation parameters and underscores the importance of simultaneous control [20].

**TCGA:** The Cancer Genome Atlas (TCGA) dataset[5] [79] represents a comprehensive and diverse collection of gene expression data derived from patients diagnosed with various types of cancer. Our research objective revolves around acquiring insights into the optimal combination of dosages for chemotherapy ($T$). In the domain of cancer treatment, the concurrent administration of different drugs has emerged as a prominent clinical approach [82]. In this context, patient survival serves as the designated outcome variable ($Y$).

We utilize the dataset as employed in previous works [5, 67]. Specifically, we use the version from [5], which includes gene expression measurements of the 4,000 genes with the highest variability, serving as the features ($X$), obtained from a cohort of $n = 9659$ patients.[6] Diverging from the approach of the aforementioned works, we once again model $T$ as a combination of different dosages that are simultaneously assigned, taking into account their joint effect. This modeling choice closely aligns with medical practice, where the optimization of drug combinations for cancer therapy necessitates a holistic consideration of each patient's unique profile [16, 82].

We summarize the characteristics of the data in Table 3.

| Variable | MIMIC | | TCGA | |
| | Dim. | Meaning | Dim. | Meaning |
|---|---|---|---|---|
| $X$ | 33 | patient covariates (age, sex, respiratory and cardiac measurements) | 4000 | gene expression measurements |
| $T$ | 2 | respiratory rate, tidal volume | 2 | chemotherapy drugs (e.g., doxorubicin, cyclophosphamide) |
| $Y$ | 1 | chance of patient survival | 1 | chance of patient survival |

Table 3: Data characteristics of our standard settings.

**Semi-synthetic data simulation:** Our data simulation process is inspired by [5]. We use the observed features $x$ from the real-world datasets. Then, for each treatment dosage $j = 1, \ldots, p$, we randomly draw two parameter vectors $v_1^{(j)}, v_2^{(j)}$ from a normal distribution $N(0, 1)$ and normalize them via $v_1^{(j)} = v_1^{(j)}/||v_1^{(j)}||$ and $v_2^{(j)} = v_2^{(j)}/||v_2^{(j)}||$, respectively. We define $\eta^{(j)} = \frac{v_1^{(j)T}x}{2v_2^{(j)T}x}$ and

---

[4]Available at: https://mimic.mit.edu
[5]Available at: https://www.cancer.gov/tcga
[6]Available at: https://github.com/ioanabica/SCIGAN

$\tilde{t}^{(j)} = \left(20 + 20 \exp(-\eta^{(j)})\right)^{-1} + 0.2$. We then sample the $j$-th dosage of the assigned dosage combination $t$ via

$$t^{(j)} \sim \text{Beta}(\tilde{\alpha}, q^{(j)}), \quad \text{where } \tilde{\alpha} = \alpha + 1, \quad q^{(j)} = \frac{\tilde{\alpha} - 1}{\tilde{t}^{(j)}} - \tilde{\alpha} + 2, \tag{12}$$

with dosage bias $\alpha = 2$, if not specified otherwise. Hence, $t^{(j)} \in [0, 1]$ can be interpreted as a percentage or normalized version of the modeled dosages.

We define the dose-response function as

$$\mu(t, x) = 2 + \frac{2}{p} \sum_{j=1}^{p} \left( \left(1 + \exp(-\eta^{(j)})\right)^{-1} + 0.5 \right) \cdot \cos\left(3\pi(t^{(j)} - \tilde{t}^{(j)})\right) - 0.01 \left(t^{(j)} - \tilde{t}^{(j)}\right)^2$$

$$- 0.1\kappa \prod_{j=1}^{p} (t^{(j)} - \tilde{t}^{(j)})^2, \tag{13}$$

where $\kappa = 1$ is an interaction parameter indicating the strength of the joint effect of the different dosages. Finally, we sample the observed outcome from $y \sim N(\mu(t, x), 0.5)$. In our data simulation, we ensure that $\tilde{t} = \left(\tilde{t}^{(j)}\right)_{j=1}^{p}$ is the optimal dosage combination and that $\tilde{t}$ additionally is the mode of the joint conditional dosage distribution of $t \mid x$ with the generalized propensity score as density given by $f(t, x) = \prod_{j=1}^{p} \text{Beta}(\tilde{\alpha}, q^{(j)})$.

Taken together, we adapt to a realistic setting in medicine where the dosage assignment is already informed to a certain degree but still can be further optimized. With increasing $\alpha$, the probability mass around the optimal dosage combinations increases, while, in areas further away from the optimum, overlap becomes more limited, and, hence, the estimation uncertainty of the dose-response function increases. We can interpolate between a random uniform distribution for $\alpha = 0$ and a completely deterministic dosage assignment for $\alpha = \infty$.

# C   Additional Results

**Robustness checks:** We now repeat the robustness checks from Sec. 5 also for the MLP baselines. The results are shown in Fig. 4 and Fig. 5. We observe that our method (DCNet+reliable) performs constantly well with respect to low regret and low (to no) variability. This holds true across the different settings. Also, we see that the baseline MLP clearly benefits from our reliable optimization during policy learning. Hence, the constrained optimization that we formulated in Steps 2 and 3 of our method are also a source of gain for existing baselines.

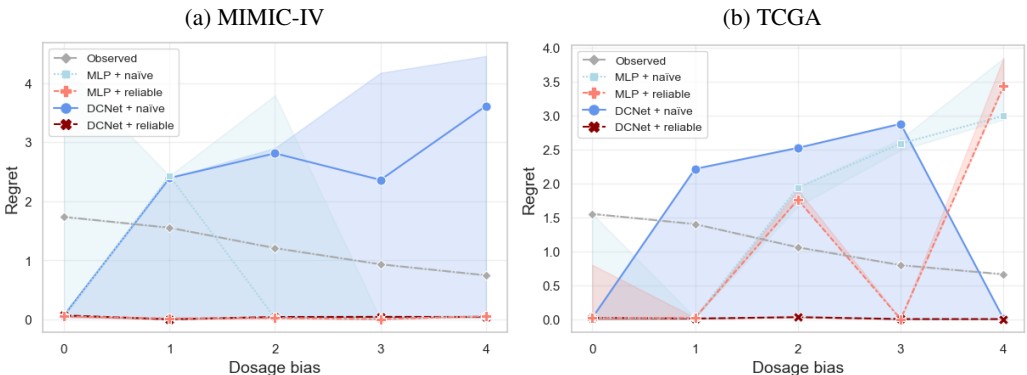

Figure 4: Robustness for dosage bias $\alpha$. *Shown:* selected policy (line) and the range over 5 runs (area). The naïve baseline has a large variability across runs while ours is highly robust as expected (i.e., we see no variability).

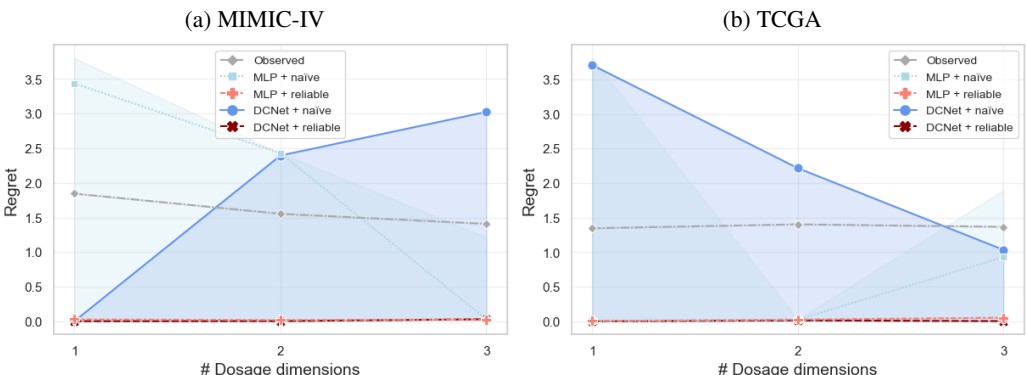

Figure 5: Robustness for number of dosages $p$. *Shown:* selected policy (line) and the range over 5 runs (area). The naïve baseline has a large variability across runs while ours is highly robust as expected (i.e., we see no variability).

**High-dimensional dosage setting:** We perform additional experiments to examine how our method scales to treatments of much higher dimensions. To improve the scalability of our method, we apply multiple prediction heads, each taking 3 treatment dimensions as input. Thereby, we fix the maximum number of jointly interacting dimensions to 3. Our motivation is that applying our method without adjustments to high treatment dimensions would lead to a high computational complexity of our prediction head because we would need to train weights for all possible interactions in the tensor product bases. However, it is often reasonable for real-world medical settings that the effective number of dimensions with joint effects is usually lower-dimensional, which is thus captured by our tailored approach. We report the results in Fig. 6. We observe that, unlike the other baselines, our method (DCNet+reliable) stays robust with low regret and low variation even for high dimensions.

**Performance across multiple re-runs:** To better assess the robustness of our method, we re-run our experiments for multiple train/val/test splits. Here, we also retrain the nuisance functions $\hat{\mu}$ and $\hat{f}$ for each split. We display the aggregated results for (i) data generated by our standard semi-synthetic

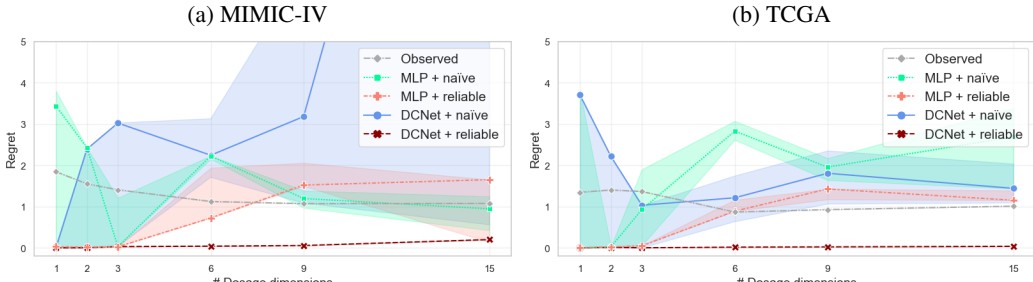

Figure 6: Robustness for high-dimensional number of dosages $p$. *Shown:* selected policy (line) and the range over 5 runs (area). Lower regret = better.

data simulation (Appendix B) in Table 4, and (ii) a more complex setting with a multimodal GPS in Table 5. For (ii), we replace the dosage combination assignment from Eq. (12) with a multimodal mixture density, i.e.,

$$t^{(j)} \sim w^{(j)} \cdot \text{Beta}(\tilde{\alpha}, q^{(j)}) + (1 - w^{(j)}) \cdot \text{Beta}(\tilde{\alpha}, u^{(j)}), \tag{14}$$

with

$$w^{(j)} \sim \text{Bernoulli}(0.5), \quad \tilde{\alpha} = \alpha + 1, \quad q^{(j)} = \frac{\tilde{\alpha} - 1}{\tilde{t}^{(j)}} - \tilde{\alpha} + 2, \quad u^{(j)} = \frac{\tilde{\alpha} - 1}{1 - \tilde{t}^{(j)}} - \tilde{\alpha} + 2.$$

| Methods | | MIMIC-IV | | | TCGA | | |
|---|---|---|---|---|---|---|---|
| $\hat{\mu}$ | $\hat{\pi}$ | selected | mean | std | selected | mean | std |
| Oracle ($\mu$) | observed | $1.089 \pm 0.067$ | - | - | $0.952 \pm 0.061$ | - | - |
| MLP | naive | $1.553 \pm 0.990$ | $1.391 \pm 0.663$ | $0.797 \pm 0.430$ | $0.832 \pm 1.067$ | $1.063 \pm 1.176$ | $0.303 \pm 0.224$ |
| MLP | reliable | $0.748 \pm 1.516$ | $0.849 \pm 1.539$ | $0.223 \pm 0.375$ | $1.080 \pm 1.542$ | $1.089 \pm 1.542$ | $0.022 \pm 0.028$ |
| VCNet | naive | $2.715 \pm 0.938$ | $2.322 \pm 0.556$ | $0.890 \pm 0.337$ | $3.169 \pm 0.588$ | $2.153 \pm 0.406$ | $0.935 \pm 0.242$ |
| VCNet | reliable | $0.033 \pm 0.027$ | $0.110 \pm 0.183$ | $0.171 \pm 0.377$ | $0.044 \pm 0.017$ | $0.114 \pm 0.145$ | $0.159 \pm 0.354$ |
| DCNET | naive | $2.512 \pm 0.787$ | $0.974 \pm 0.700$ | $1.134 \pm 0.336$ | $2.048 \pm 0.775$ | $1.682 \pm 0.478$ | $0.776 \pm 0.263$ |
| DCNET | reliable (**ours**) | $0.025 \pm 0.021$ | $0.024 \pm 0.020$ | $0.001 \pm 0.001$ | $0.025 \pm 0.010$ | $0.263 \pm 0.540$ | $0.337 \pm 0.750$ |

Reported: selected / mean / standard deviation (lower = better)

Table 4: Regret of 5 re-runs ($mean \pm std$) with different train/val/test splits with each $k = 5$ restarts for policy learning.

| Methods | | MIMIC-IV | | | TCGA | | |
|---|---|---|---|---|---|---|---|
| $\hat{\mu}$ | $\hat{\pi}$ | selected | mean | std | selected | mean | std |
| Oracle ($\mu$) | observed | $1.294 \pm 0.156$ | - | - | $1.624 \pm 0.071$ | - | - |
| MLP | naive | $2.139 \pm 1.539$ | $0.860 \pm 0.566$ | $0.993 \pm 0.447$ | $1.745 \pm 1.270$ | $1.444 \pm 1.432$ | $0.640 \pm 0.166$ |
| MLP | reliable | $1.069 \pm 1.518$ | $0.544 \pm 0.509$ | $0.786 \pm 0.656$ | $0.465 \pm 0.902$ | $0.720 \pm 1.488$ | $0.157 \pm 0.328$ |
| VCNet | naive | $1.885 \pm 0.258$ | $1.627 \pm 0.149$ | $0.443 \pm 0.152$ | $1.728 \pm 0.306$ | $1.659 \pm 0.176$ | $0.368 \pm 0.130$ |
| VCNet | reliable | $0.126 \pm 0.065$ | $0.192 \pm 0.140$ | $0.147 \pm 0.193$ | $0.029 \pm 0.027$ | $0.119 \pm 0.117$ | $0.201 \pm 0.279$ |
| DCNET | naive | $0.713 \pm 0.801$ | $0.389 \pm 0.256$ | $0.515 \pm 0.347$ | $0.260 \pm 0.409$ | $0.141 \pm 0.125$ | $0.209 \pm 0.194$ |
| DCNET | reliable (**ours**) | $0.077 \pm 0.059$ | $0.065 \pm 0.038$ | $0.061 \pm 0.058$ | $0.014 \pm 0.010$ | $0.028 \pm 0.026$ | $0.031 \pm 0.060$ |

Reported: selected / mean / standard deviation (lower = better)

Table 5: Regret of 5 re-runs ($mean \pm std$) with different train/val/test splits with each $k = 5$ restarts for policy learning in an additional setting simulated with a **multimodal GPS**.

We can interpret the results as follows:

(1) The column "selected" displays the final performance of our method when using the finally recommended policy. In this column, the $mean \pm std$ can be considered as the expected regret with Monte-Carlo-CV confidence intervals (including retraining the nuisance functions) and should primarily be used for evaluating the performance and for comparison with different policy learning methods. We observe that our method significantly outperforms the baselines.

(2) The column "mean" represents the means within the $k$ restarts of the policy learning (i.e., Step 3 in our method). As such, its values can indicate how a randomly selected policy out of the $k$ restarts would perform compared to the "selected one". For example, we observe that, for the "naïve" baselines using a naïve MSE loss for selection, suboptimal policies are oftentimes selected, whereas our method shows robust behavior.

(3) The column "std" shows the standard deviation within the $k$ restarts of the policy learning (i.e., Step 3 in our method). As such, the values indicate how often similar performing policies are learned. It thus refers to the robustness within the $k$ runs. As a consequence, lower values imply that (i) probably fewer restarts $k$ are needed to find the targeted optimum, and (ii) fewer local optima are learned, which reduces the risk of selecting a suboptimal policy.

Hence, we suggest using the "selected" column ($mean \pm std$) for evaluating the final performance of our method, and the other two columns ("mean" and "std") to show the inner robustness of our method compared to ablation baselines.

Overall, we demonstrate that the performance of our framework is robust across the tested settings and has low regret and low variation across different re-reruns and different restarts, including the more complex setup.

**Sensitivity analysis of the reliability threshold $\bar{\varepsilon}$:** In our experiments, we set the reliability threshold $\bar{\varepsilon}$ to the $5\%$-quantile of the estimated GPS $\hat{f}(t, x)$ of the train set. This is due to the reason that the GPS is a probability density function over a $p$-dimensional continuous space and, hence, is hard to interpret intuitively. Unlike discrete probabilities, which are bounded between $0$ and $1$ and can be interpreted intuitively by humans, the GPS has no upper bound. To this end, we suggest a heuristic to choose $\bar{\varepsilon}$ dependent on $x\%$-quantiles of the estimated GPS based on the train dataset. As a result, we aim to learn optimal dosages, which have at least the same estimated reliability as $x\%$ of the observed data.

In our heuristic, the selection of the quantile $x$ is still subjective and cannot be optimized during hyperparameter tuning, as we assume to have no access to the ground truth $\mu(t, x)$. Nevertheless, we can leverage our semi-synthetic data to perform a sensitivity analysis over $x$ for $\bar{\varepsilon}$ and thus corroborate our choice. For this, we evaluate the final performance of our method on the test data for $\bar{\varepsilon}$ set to different quantiles of $\hat{f}$ over $k = 5$ runs. The results are displayed in Fig. 7.

We observe that, depending on the dataset, different quantiles for selecting $\bar{\varepsilon}$ can lead to differences in variability. However, we find that lower values tend to lead to less variability. Also, we find that, even when there is higher variability within the $k$ runs, for every setting, a run with low regret is selected. We attribute this to our validation loss from Eq. 9. Together, the results are in line with our expectations and add to the robustness of our methods.

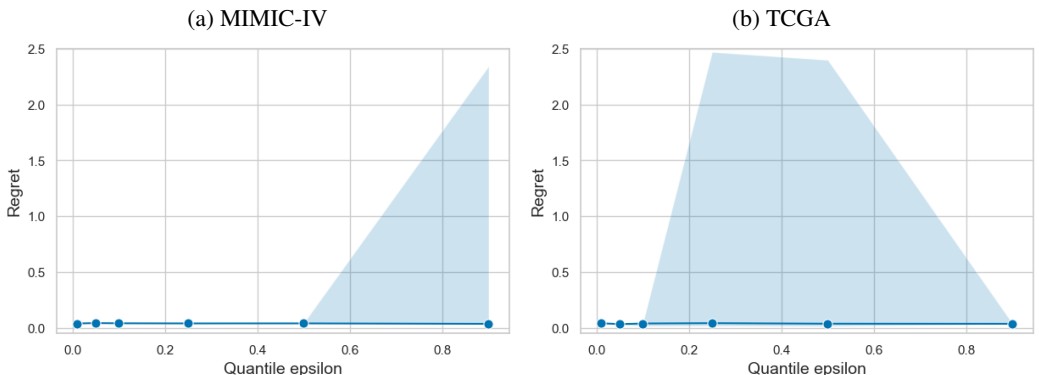

Figure 7: Sensitivity analysis of the reliability threshold $\bar{\varepsilon}$ for different quantiles of $\hat{f}$ of the train set. *Shown:* selected policy (line) and the range over 5 runs (area).

# D   Insights

**Individualized dose-response estimates:** In our experiments in Sec. 5 and in Appendix C, we showed a comparison to naïve baselines. Therein, in particular, we mainly demonstrated that the reliable optimization step in our method is an important source of performance gain. We now give additional intuition as to why also our DCNet is a source of performance gain. In the following, we first demonstrate that our DCNet is able to estimate the dose-response function more accurately than the baseline in regions with sufficient overlap. We then discuss why this brings large gains for our reliable optimization. Thereby, we offer an explanation why both DCNet and reliable policy optimization are highly effective when used in combination.

In Table 6, we display the performance of the dose-response estimators. In detail, we report: (i) the mean squared error ("Biased MSE"), calculated on a holdout test dataset which follows the same biased dosage assignment as the train data; (ii) the mean ("MISE", see e.g., [67]) of the integrated squared error calculated on areas with sufficient overlap, i.e. $\sum_{i=1}^{n} \int_{[0,1]^p} (\mu(t, x_i) - \hat{\mu}(t, x_i))^2 dt \mid \hat{f}(t, x) > \bar{\varepsilon}$.; and (iii) the standard deviation of the integrated squared error on areas with sufficient overlap ("SD-ISE"). Intuitively, the MISE and SD-ISE are the expected value and variation of the prediction error of the dose-response estimator over the covariate-treatment space with sufficient overlap. Hence, a low value implies that all suitable dosage combinations with sufficient overlap are estimated properly and that certain, less frequent or non-dominant combinations are not neglected. We observe that our DCNet also performs clearly best wrt. to all metrics, undermining its ability to successfully account for the different drug-drug interactions under dosage bias.

| Methods | MIMIC-IV | | | TCGA | | |
|---|---|---|---|---|---|---|
| $\hat{\mu}$ | Biased MSE | MISE | SD-ISE | Biased MSE | MISE | SD-ISE |
| MLP | 0.293 | 0.049 | 0.069 | 0.438 | 0.374 | 0.745 |
| VCNet | 0.691 | 0.558 | 0.749 | 0.551 | 0.406 | 0.474 |
| **DCNet** (ours) | 0.260 | 0.013 | 0.032 | 0.257 | 0.006 | 0.010 |

Table 6: Estimation error of dose-response estimators.

To give further intuition to the above, we select one random observation per dataset and plot the following functions: (i) the estimated GPS, (ii) the oracle individualized dose-response surface, (iii) the estimates using DCNet, and (iv) the estimates using the baseline MLP. We compare two settings, namely with no dosage bias ($\alpha = 0$) and thus sufficient overlap (in Fig. 8) and with dosage bias ($\alpha = 2$) and thus clearly limited overlap (in Fig. 9).

The plots can be interpreted as follows. With no dosage bias (Fig. 8), both DCNet and MLP seem to approximate $\mu$ adequately (with slightly better estimates of DCNet at the modes at the boundary areas) and are thus suitable for naïve policy learning. However, the findings change in a setting with limited overlap (Fig. 9). Here, the limited overlap is indicated by areas with low estimated GPS $\hat{f}$. Both estimators for $\hat{\mu}$ show clear deviations to the oracle $\mu$. However, the deviations are smaller for large parts in the case of our DCNet (as compared to the MLP). Evidently, the dose-response surface predicted by the MLP loses its expressiveness, especially for the high-dimensional TCGA dataset, and fails to model the maximum adequately. This leads to erroneous predictions when used as a plug-in estimator for policy learning, which can not be compensated by constraining the search space to regions with sufficient overlap. DCNet, on the other hand, is able to model the dose-response surface more adequately in areas with sufficient overlap due to its custom prediction head ensuring expressiveness. However, in regions with limited overlap, the splines of the prediction head extrapolate and lead to strongly incorrect predictions. Anyway, as our goal is to learn a reliable policy, we aim to exclude the regions with limited overlap and thus high uncertainty (yellow regions) such that they do not affect our final predicted policy.

In conclusion, when now applying our reliable policy optimization on top of our DCNet, we constrain the search space to the reliable regions, in which DCNet outperforms the MLP. Thus, we ensure reliable and accurate predictions for policy learning by using our method.

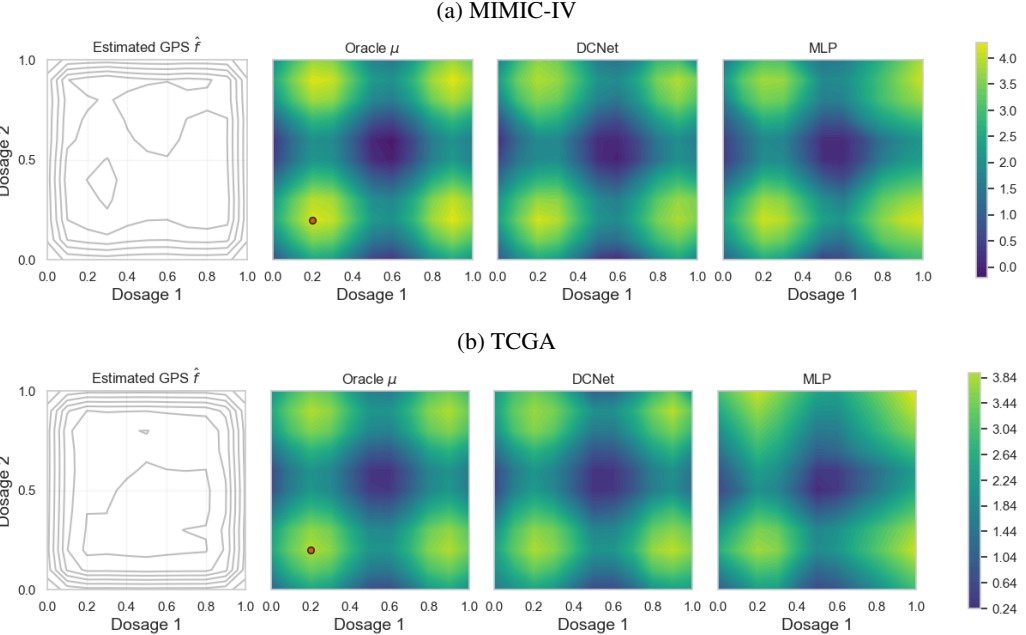

Figure 8: Insights for one randomly sampled observation in the setting with dosage bias $\alpha = 0$. We show (i) the estimated GPS and, additionally, the individualized dose-response surface through (ii) the true oracle function, (iii) the estimates using our DCNet, and (iv) the estimates using the baseline MLP. The optimal dosage combination is shown as a red point.

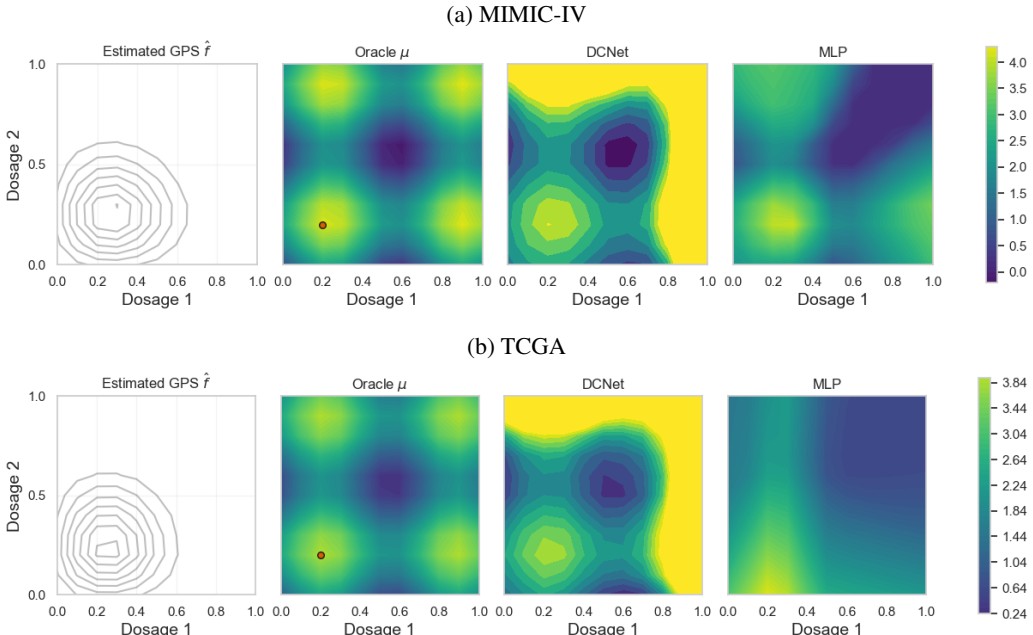

Figure 9: Insights for one randomly sampled observation in the setting with dosage bias $\alpha = 2$. We show (i) the estimated GPS and, additionally, the individualized dose-response surface through (ii) the true oracle function, (iii) the estimates using our DCNet, and (iv) the estimates using the baseline MLP. The optimal dosage combination is shown as a red point.

**Computational complexity of policy optimization:** To ensure scalability at inference time, we train an additional policy neural network to directly predict the optimal policy, instead of performing optimization for new patients. We explain our motivation in the following: The complexity of a direct optimization at inference time is non-trivial and depends on the complexity of the forward pass of DCNet $\sigma_\mu$ and the CNFs $\sigma_f$, as well as the selected constrained optimization algorithm, and the number of patients to evaluate $n_{\mathrm{eval}}$. While the exact complexity of the direct optimization depends on the specific solver cost $\sigma_{\mathrm{solve}}$, which, for most algorithms, scales heavily with the dosage dimension $p$, it is at least $\mathcal{O}\left(n_{\mathrm{eval}} \times (\sigma_\mu + \sigma_f) \times \sigma_{\mathrm{solve}}\right)$. This becomes costly for large-scale datasets in practice. In contrast, our policy network is trained only once. Hence, upon deployment, it simply needs operations for evaluation, which is at most $\mathcal{O}(n_{\mathrm{eval}} \times \sigma_{\mathrm{net}})$, where $\sigma_{\mathrm{net}}$ is the complexity of only the forward pass of the policy network. Hence, it usually holds $\sigma_{\mathrm{net}} \ll \sigma_{\mathrm{solve}}$.

# E   Modeling and Training Details

**Dose-response estimator:** For training our DCNet $\hat{\mu}(t, x)$, we follow previous works from the baseline VCNet [2, 56]. We set the representation network $\phi$ to a multi-layer perceptron (MLP) with two hidden layers with 50 hidden neurons each and ReLU activation function. For the parameters of the prediction head $h$, we use the same model choices as for $\phi$. Additionally, we use B-splines with degree two and place the knots of the tensor product basis at $\{1/3, 2/3\}^p$. For the baseline MLP from Sec. 5, we ensure similar flexibility for fair comparison, and we thus select a MLP with four hidden layers with 50 hidden units each and ReLU activation. We train the networks minimizing the mean squared error (MSE) loss $\mathcal{L}_\mu = \frac{1}{n} \sum_{i=1}^n \left( \hat{\mu}(t_i, x_i) - y_i \right)^2$. For optimization, we use the Adam optimizer [43] with batch size $1000$ and train the network for a maximum of $800$ epochs using early stopping with a patience of $50$ on the MSE loss on the factual validation dataset.

We tune the learning rate within the search space $\{0.0001, 0.0005, 0.001, 0.005, 0.01\}$, and, for evaluation, we use the same criterion as for early stopping.

**Conditional normalizing flows:** For modeling the GPS $\hat{f}(t, x)$, we use conditional normalizing flows (CNFs). Specifically, we use neural spline flows [15] in combination with masked auto-regressive networks [13]. We use a flow length of $1$ with $5$ equally spaced bins and quadratic splines. For the autoregressive network, we use a MLP with 2 hidden layers and 50 neurons each. Additionally, we use noise regularization with noise sampled from $N(0, 0.1)$. We train the CNFs by minimizing the negative log-likelihood (NLL) loss $\mathcal{L}_f = -\frac{1}{n} \sum_{i=1}^n \log \hat{f}(t_i, x_i)$. For optimization, we use the Adam optimizer with batch size $512$ and train the CNFs for a maximum of $800$ epochs using early stopping with a patience of $50$ on the NLL loss on the factual validation dataset.

We tune the learning rate within the search space $\{0.0001, 0.0005, 0.001, 0.005, 0.01\}$ and for evaluation we use the same criterion as for early stopping.

**Policy learning:** After training the nuisance models $\hat{\mu}$ and $\hat{f}$, we use them as plug-in estimators for our policy learning. For the policy network $\hat{\pi}_\theta$, we select a MLP with 2 hidden layers with 50 neurons each and ReLU activation. We set the reliability threshold $\bar{\varepsilon}$ to the $5\%$-quantile of the estimated GPS $\hat{f}(t, x)$ of the train set, when not specified otherwise. We use Adam optimizers for updating $\theta$ and $\lambda$ each, with batch size $512$, and train the network using the gradient descent-ascent optimization objective from Eq. (7) for a maximum of $400$ epochs using early stopping with a patience of $20$ on the validation loss from Eq. (9) on the factual validation dataset.

We set the learning rate for updating $\lambda$ to $\eta_\lambda = 0.01$ and use random search with 10 configurations to tune the learning rate for updating the parameters of the policy network $\eta_\theta \in \{0.0001, 0.0005, 0.001, 0.005, 0.01\}$ and the initial value of the Lagrangian multipliers $\lambda_i \in [1, 5]$. To evaluate the performance during hyperparameter search, we use the same criterion as for early stopping. Then, after determining the hyperparameters, we perform $k = 5$ runs to find the optimal policy.

Our above procedure was chosen to ensure a fair and direct comparison to previous literature and to the naïve baselines, which enables us to show the sources of performance gain in our method (see Sec. 5). Note that when applying our method for reliable policy learning for dosage combinations in production, one might consider optimizing more exhaustively also over the different modeling choices of the nuisance estimators and the policy network.

