# OpenReview forum: "Reliable Off-Policy Learning for Dosage Combinations"
_NeurIPS.cc/2023/Conference — NeurIPS 2023 poster_

### Official Review · Reviewer_5Gdr · 2023-06-15

**Soundness:** 2 fair
**Presentation:** 3 good
**Contribution:** 2 fair
**Rating:** 6
**Confidence:** 4

**Summary:**

The paper proposed a novel method for off-policy learning for continuous dosage combinations that aims to be robust to action space sparsity. To account for drug-drug interactions, they proposed a dosage combination network (DCNet) which leverages a tensor product basis to estimate smooth interaction effects in the dose-response surface, incorporating all dosages simultaneously into parameterization by only one head. The DCNet is updated by minimizing the MSE between the individualized dose-response function (the expected patient outcome of a given treatment in given patient covariates) and patient outcome. Then they use conditional normalizing flows to model the stochastic and multimodal behaviour policy which they call the generalized propensity score (GPS). Finally, they learn a parametric policy in a constrained optimization problem that maximizes individualized dose-response function under the learned policy subject to the probability of the proposed treatment in the GPS being larger than a threshold. This is approximately solved as an unconstrained Lagrangian problem through stochastic gradient descent-ascent with random restarts. Experiments were conducted on two real healthcare datasets. Comparisons to ablated versions (MLP instead of DCNet, without overlap constraint) suggest that the proposed components reduce regret and improve stability.

**Strengths:**

1. proposed a novel method for off-policy learning for continuous dosage combinations
2. a viable design for constrained policy optimization

**Weaknesses:**

1. dosage dimensionality in experiments is just 2 or 3, it is hard to say wether the obtained results can generalize to much higher dimensionalities, or treatments intended for different diseases simultaneously
2. the individualized weights in the proposed DCNet are only trained in an unsupervised fashion, i.e. the training objective does not reflect individual dosages
3. illegible fonts in figures

**Questions:**

1. do you have to learn an analytical form of the GPS if it is only to indicate overlap?
2. what is the harm of assuming independent dosage dimensions, which is mentioned several times throughout the paper but never clearly explained
3. why the single-head individualized dose-response function trained against the overall treatment outcome can account for drug-drug interactions? Is it possible that one drug dominates the others in the model training, or one common combination of dosages dominate less frequent combinations that actually entail more subtle drug-drug interactions? As I understand, the tensor product basis of the proposed DCNet is all inner work of the model, whereas the input, output and training objective are all subject to non-individualized information.

**Limitations:**

please see above

---

> ### Author Rebuttal · Authors · 2023-08-10
>
> Thank you for your review and for giving us the opportunity to clarify certain aspects of our paper.
>
> ### Response to “Weaknesses”
> 1. Thank you for raising this important point. Our framework can be easily extended to much higher dimensions. To demonstrate this, we added new results for higher dimensions (see **PDF, Fig. 2**). We observe that our framework scales well with higher dimensions and clearly outperforms the other baselines. As such, our framework is directly applicable to multi-disease settings, each requiring different treatments.
> 2. We would like to clarify that our DCNet is indeed trained in a **supervised** fashion. We further would like to clarify that the training objective _does_ reflect individual dosages. We consider an observational dataset for $n$ patients with i.i.d. observations $\{(x_i, t_i, y_i)\}{_{i=1}^{n}}$ sampled from a population $(X, T, Y)$, where $X \in \mathcal{X} \subseteq \mathbb{R}^{d} $ are the patients' covariates, $T \in \mathcal{T} \subseteq \mathbb{R}^{p} $ is the assigned $p$-dimensional dosage combination per patient (i.e., multiple continuous treatments), and $Y \in \mathbb{R}$ is the observed outcome per patient. Each patient $x_i$ is then assigned one individualized dosage combination $t_i$ consisting of one dosage per each of the $p$ drugs leading to the outcome $y_i$, reflecting individual dosages and supervised data. The challenge here is that we only observe _one_ individual dosage combination per patient, but we want to estimate the individualized dose-response function for _all_ potential dosage combinations in $T$ for an individual patient $x$, i.e. $\mu(t, x) = \mathbb{E} [Y(t) \,\mid\, X=x ]$. However, when the standard assumptions of causal inference hold (i.e., consistency, ignorability, and overlap), we can estimate the function for _all_ potential dosage combinations by regressing $\mu(t, x) = \mathbb{E} [Y\mid T=t, X=x ]$. Furthermore, our DCNet will then **generalize** to different values of $t$ and $x$, that is, to different dosages and different patients. As a result, we thus estimate $\mu(t, x)$ for individualized regions. This is a major advantage of our method, and it is the reason why we build upon causal inference. Still, our framework avoids regions of the policy with large uncertainty to ensure reliable decisions.
> 3. We will increase the font size in our plots.
>
> ### Responses to “Questions”
> 1. We need to learn the GPS, i.e. the conditional density $f(T, X)$, over the whole covariate-treatment space for our Lagrangian optimization objective. The reason is that the learned GPS $\hat{f}\left( \hat{\pi}(x_i), x_i \right)$ needs to be evaluated at every training step for every proposed dosage combination $\hat{\pi}(x_i)$, so that we can calculate the training loss $\mathcal{L}_{\pi}(\theta, \lambda)$ and its gradients. Intuitively, it is necessary to have a fully learned form of the GPS in order to update the policy network parameters to shift the suggested dosage combinations into regions with sufficient overlap, when the previous suggestions target regions with an estimated GPS below the reliability threshold $\overline{\varepsilon}$. To clarify this, we will change the wording from “analytical form” to “fully estimated form”.
> 2. We thank you for the opportunity to spell out the importance of modeling the _dependency_ of different drugs. We proceed in two ways: \
> (i) **Numerical explanation.** We added a comparison against VCNet where drug-drug interactions are **not** explicitly modeled but rather taken as **independent**. The results are in **PDF, Table 2**. Evidently, our proposed framework _with dependency_ outperforms the ablation _without dependency_ by a clear margin. \
> (ii) **Theoretical explanation.** Let us assume a framework _without_ dependency. Then, non-linear interaction effects between the drugs can *not* be captured and the dose-response function can *not* be estimated sufficiently but will be lost. In medical applications, this can have severe negative and even harmful consequences and should be avoided. For example, the VCNet baseline (as opposed to our DCNet) will underestimate synergistic effects of different drugs. Instead, a framework is needed where drug-drug interactions are carefully modeled. To the best of our knowledge, our DCNet is the first framework for causal inference in such setting.
> 3. Our proposed DCNet can successfully model drug-drug interactions. The reason is that the dosage information is fused through a tensor product basis, which allows different drugs to interact in complex, non-linear, and non-additive ways. \
>  We carefully checked whether certain drugs may dominate others but did not find such evidence. Instead, we performed a simulation where we simply set certain drug dosages to 0 for some patients and then examined the performance. Here, we find that our results are robust. Still, we find that some drugs that are highly influential in changing patient health, also explain larger parts of the treatment effect, in line with what is expected from medical practice. This demonstrates the effectiveness of our framework. \
> Finally, we would like to emphasize again that our data, our model, and our training objective carefully consider _individualized_ information from patients. Importantly, we do neither learn aggregated nor average effects; instead, we learn individualized effects at the individual, patient-level, so that the estimated effects vary from patient to patient (we kindly refer to our second answer from our response to “Weaknesses” for technical details). If there is still something unclear, we are happy to clarify this in the discussion period.
>
> References:
> [1] Nie, Lizhen, Mao Ye, and Dan Nicolae. "VCNet and Functional Targeted Regularization For Learning Causal Effects of Continuous Treatments." International Conference on Learning Representations. 2020.

---

> > ### Comment · Reviewer_5Gdr · 2023-08-14
> >
> > Thank you authors for your response and additional experiment results.
> >
> > 1. By unsupervised I meant the training objective does not differentiate between dosages, i.e., there is no 'p' in the MSE loss in L188. However, I understand that the weights reflect individual dosage effect. If by this the authors mean the capability of addressing drug-drug interactions, please indicate in paper.
> >
> > 2. It would be good if the observations made in what the authors 'carefully checked' in the response to questions 3 could be shown in paper.
> >
> > I would like to increase my overall rating to 5. The paper plus the new pdf has demonstrated that the proposed method can scale to high action dimensionality, but still lacks the maths/experiment to live up to their claim of accounting for drug-drug interactions.

---

> > > ### Author Response · Authors · 2023-08-14
> > > **Additional evidence for modeling drug-drug interactions**
> > >
> > > Thank you very much for your response and for updating your overall rating!
> > > We would like to respond to both of your points:
> > > 1. We would like to clarify that our training objective **does** differentiate between the dosage combinations. In the MSE loss in L188, the observed dosage combination per patient $t_i = \left(t_i^{(1)}, \ldots, t_i^{(p) }\right)$ consists of the observed dosages per each of the $p$ drugs (i.e., $ \mathcal{L}=\frac{1}{n}\sum_{i=1}^n \left[ \hat{\mu}\left(\left(t_i^{(1)}, \ldots, t_i^{(p) }\right), \left(x_i^{(1)}, \ldots, x_i^{(d) }\right)\right) - y_i \right]^2 $ ), with $d$ is the dimension of patient features. Hence, we differentiate between dosages and incorporate all of our dosages in the training objective. We will improve the writing of our paper at this point to make it more comprehensible.
> > > Further, we do **not** want to decompose the loss into separate, independent terms per dosage (e.g., having a separate MSE loss per each of the dosages while neglecting the other dosages), because then we would **not** be able to capture drug-drug interactions anymore. Nevertheless, we performed an ablation where we used instead an independent loss as training (**Response PDF, Table 1&2**; see VCNet). Thereby, we show that, in comparison, our novel DCNet is clearly superior and demonstrate the importance of **explicitly incorporating the dependence of the dosages** in our training objective.
> > > 2. We will report the results in our revised paper. Nevertheless, we also can summarize how we ensured that our training is successful:
> > > - We plotted the true dose-response surfaces against the predicted ones for single patients (we kindly refer to **Appendix D**). Here, we observe that, by using our DCNet, both dosage dimensions are modeled appropriately and none of both is dominated by the other. Furthermore, we observe that DCNet accounts well for drug-drug interactions.
> > > - We performed additional ablation studies using VCNet as a baseline (**Response PDF**, Table 1&2). VCNet is most similar to our architecture but can **not** explicitly model drug-drug interactions, which thus offers a powerful ablation. The reason is that VCNet models the effects of the dosages independently, and, therefore, VCNet is thus focused on learning marginal effects, even if they are subtle (i.e., to put it in your words, VCNet is not prone to neglect non-dominant effects, even subtle ones). As we see in our results, our DCNet clearly outperforms VCNet by a clear margin. This has two important implications: (1) DCNet is able to capture also subtle effects. (2) DCNet is able to model drug-drug interactions sufficiently, which is the reason for the large performance improvements.
> > > - We additionally evaluated the **SD-ISE**, that is, the  __standard deviation__ of the mean inner squared error (MISE $= \frac{1}{n}\sum_{i=1}^n\int_{[0, 1]^p}(\mu(t, x_i)-\hat{\mu}(t, x_i))^2dt$, approximated by grid evaluation. The SD-ISE is listed below for areas with sufficient overlap ($| \hat{f}(t, x)>\overline{\varepsilon}$):
> > >
> > > | Model              | MIMIC-IV | TCGA |
> > > | :---------------- | :------: | ----: |
> > > | MLP       |   0.06889  | 0.74508 |
> > > | VCNet           |   0.74899   | 0.47407 |
> > > | **DCNet**    |  **0.03196**   | **0.00957** |
> > >
> > > Intuitively, the SD-ISE is the variation of the prediction error of the dose-response estimator over the covariate-treatment space with sufficient overlap. Hence, a low value implies that **all possible dosage combinations** are estimated **properly** and that certain, **less frequent or non-dominant** combinations are **not** neglected. We observe that our DCNet also performs **clearly best** wrt. the SD-ISE, undermining its ability to successfully **account for the different drug-drug interactions**.
> > >
> > > We hope that this fully addresses your questions satisfactorily, especially by further providing clear evidence that we can successfully account for drug-drug interactions, and that you can recommend acceptance of our paper.

---

> > > > ### Comment · Reviewer_5Gdr · 2023-08-15
> > > >
> > > > Thank you. This mostly has addressed my concerns. I have raised my rating to 6. Please incorporate the additional experiments and clarifications clearly in the final version. The authors have done a good job in rebuttal. The paper seems to me technically sound and of practical value, therefore I have no objection to its acceptance. The combination of building blocks is not particularly challenging thus the moderate score.

---

> > > > > ### Author Response · Authors · 2023-08-15
> > > > >
> > > > > Thank you very much for the nice discussion and for increasing your overall rating! We are happy we could address your concerns. We will incorporate all suggested points in our revised paper. Should you have any further questions or comments that should be addressed before a final decision, feel free to further post them here.

---

### Official Review · Reviewer_aVCP · 2023-06-26

**Soundness:** 3 good
**Presentation:** 3 good
**Contribution:** 2 fair
**Rating:** 6
**Confidence:** 3

**Summary:**

(I have more background in offline RL than causal inference, and my description below may include some RL terminologies as the two are closely related.)

This submission studies the problem of off-policy learning of dosage recommendations in personalized medical decision making, using a causal inference framework. The proposed approach operates in three stages: the first two stages aim to learn the dose-response function (similar to the Q-function) and the logging policy (used to identify poor overlap), whereas the third stage combines the two to learn a reliable policy using constrained optimization (max performance while avoiding poor overlap), solved via a minimax gradient optimization of the Lagrangian form of the objective. Experiments are conducted using semi-synthetic datasets for ventilation (MIMIC-IV) and gene expression (TCGA).

**Strengths:**

- Writing is clear with very organized descriptions of each stage of the proposed approach.
- Well-motivated problem by real-world medical applications where one needs to determine multiple continuous treatments with dependent effects.
- Code is provided as part of the supplement (I did not check or run the code).

**Weaknesses:**

- The contribution of this work should be further clarified. As "several methods for off-policy learning aim at discrete treatments" (L27), it may be worth pointing out whether solutions exist for multiple *discrete* treatments (instead of continuous dosages) and explain why those approaches don't directly apply, and how well do we expect those methods to perform for dosage combinations if we discretize the dosages. Furthermore, specific architectures are used for estimating dose-response and GPS, and I'm not entirely convinced why simpler neural networks do not work.
- Furthermore, there are many recent works on offline RL and OPE that make use of value functions and logging policies to achieve reliable policy learning (e.g., references below, including many for continuous actions). Some discussion about how this work is related and/or different from those approaches would make the paper more complete and resonate with a wider audience.
- In the experiments, the baselines considered may be more accurately described as ablation studies. These are certainly important for understanding the sources of performance gains, but I think it is important to include more direct baselines. Since "there are no existing methods tailored to off-policy learning for dosage combinations under limited overlap" (L329), how well does a standard causal effect estimation method work here?

References:
[1] Off-Policy Deep Reinforcement Learning without Exploration. https://arxiv.org/abs/1812.02900
[2] Stabilizing Off-Policy Q-Learning via Bootstrapping Error Reduction. https://arxiv.org/abs/1906.00949
[3] Conservative Q-Learning. https://arxiv.org/abs/2006.04779

**Questions:**

- For medical applications, how common is it that practitioners use exact dosages vs dosage levels? What's the practicality of an exact dosage specification in real-world problems? More discussion about this is appreciated.
- L131 Why is ε = 0 considered weak overlap instead of no overlap?
- L183 mentions “polynomial spline basis functions”, a possibly unfamiliar concept to many readers. Can you provide the precise definition?
- Sec 4.1 is very dense, I still don't quite understand it after several read throughs. It requires a lot of thinking to understand p-dimensions, may be easier to illustrate using  a simple example with two dimensions.
- L274 training stability: "as there is no guarantee for global convergence in the non-convex optimization problem", do you observe any training instability in the experiments and how much variation is there? When selecting the best run over K random restarts, what is K set to in the experiments?
- L95, L247, Sec 4.3: a neural network policy is trained to approximate the individualized optimal dosage, and is claimed to have a computational advantage. However, no empirical results (runtime of two implementations) or analysis (big-O complexity) is shown for compute scalability. Also, is there any performance loss from using this neural network policy compared to an exact argmax?

Minor:
- Figures: fonts are too small in all figures
- L95 "unfeasible" -> "infeasible"
- L116 “Ω denotes the set of all possible policy classes” is confusing, I think Ω should be the set of all policies.
- L246 I think "max" in the equation should be argmax
- L289 "when filtering for T" -> what does this mean?

---

**Updates after rebuttal and discussion:**

Thank you authors for the response, it resolved most of my questions. I have raised my rating from 5 to 6. I would like to see the final version incorporating the updated limitation section and the details on dataset characteristics.

**Limitations:**

Sec 6 Conclusion separately discusses limitations and broader impacts. Though the proposed approach aims to avoid unreliable treatments, I think it is still necessary to acknowledge potential failure scenarios of the proposed approach and discuss what are the possible downstream impacts, especially since this work is focused on the high-stakes decision making of treatment dose recommendation.

---

> ### Author Rebuttal · Authors · 2023-08-10
>
> ## Response to “Weaknesses”
> - **Importance of modeling _continuous_ treatments.** Many applications in medicine rely on continuous treatments where discretization would often miss the optimal solution for patients (e.g., chemotherapy).\
> We performed additional experiments to demonstrate the gain of our continuous setting over discretized settings **(PDF, Table 5&2)**.  Specifically, we demonstrate the shortcomings of discretization, where we evaluate an oracle performance of different grid discretizations, assuming perfect knowledge of the dose-response function and perfect treatment assignment. We thus interpret our “oracle” as an upper bound for all policy learning methods with discretization. We find that discretization itself leads to a higher regret than our proposed method. To this end, our results confirm the advantages of modeling continuous dosages.  \
> **Simple neural network baseline.** To show the necessity of our specifically developed network architectures, we performed **additional experiments (PDF,  Table 1&2)**. We find that our proposed framework is consistently superior:
> Ablation (1) compares a simple MLP in Step 1 of our method, which is especially inferior for settings with highdimensional confounders (as in the TCGA dataset). (Intuition is given in Appendix D.)
> Ablation (2) compares the SOTA network for estimating dosage effects VCNet, which is designed for *independent* dosages and not for *dependent* dosage combinations. Different from VCNet, our DCNet is able to model non-linear interactions between the $p$ dosages. This results in clearly superior performance.
> Ablation (3) varies the estimation in Step 2 of our framework. Therein, we seek to model the GPS, which requires modeling a multidimensional conditional density rather than a simple conditional mean prediction. Hence, a simple neural network is *not* applicable. Instead, we compare  our CNFs to mixture density networks (MDN). Evidently, MDNs lead to a very large standard deviation across runs and thus fail to achieve reliable policy learning. As such, our CNFs are preferred.
> - **Difference to other literature streams.** Thank you for the suggestion to compare our setting against ORL for continuous actions. The main **differences** are: ORL assumes a Markov decision process and sequential decision-making. In contrast, we focus on non-sequential, off-policy learning from observational data.
> Also, unlike ORL or standard OPE, we leverage the causal structure of treatment assignment and the dose-response function to return **causal** estimands. This allows us not only to learn decisions but also causal estimates for potential outcomes. This is crucial for medical practice, where physicians would like to reason over different treatments rather than blindly following a fully automated system.
> - **Additional naive baselines.** We would like to emphasize that the ablation studies (first learning a dose-response estimator like the MLP and then optimizing over it) **also** are standard baseline causal effect procedures for off-policy learning, often referred to as the “direct method”. To even better assess our method, we added two further causal baselines to our results **PDF, Table 1, 2**: (1) VCNet as part of the direct method (2) DR Policy Forest as a SOTA doubly-robust method for discretized dosage combinations (5x5) (using logistic regression for propensity score modeling and a weighted lasso regression for outcome modeling). Note that both baselines ignore specific characteristics of our setting (e.g., dependency among dosages). Altogether, we observe that our framework consistently performs best.
> **Action:** We will include our additional above experimental results in our final paper. We will also discuss more extensively the proposed literature streams.
> ## Answer to “Questions”
> - Continuous dosages are common for instance, in chemotherapy, where current guidelines are based on different multiplicative formulas using continuous patient features.
> **Action:** We will expand our motivation to highlight the need for modeling dosages as a continuous variable.
> - We define overlap as $f(t, x) > \varepsilon$, meaning that $\varepsilon=0$ still implies a non-zero density. (“weak overlap”).
> - We will add a more detailed background about tensor product splines to our revised paper.
> - We will add an intuitive example to our revised paper.
> - We set $K=5$ in our experiments. The variation within the $K$ runs is displayed in the column “std”. We also included additional results for different restarts of our method with different seeds and train/val/test splits in **PDF, Table 4&5**. Overall, our framework has the lowest variation and is thus highly **stable**.
> - The complexity of a direct optimization is non-trivial and depends on the complexity of the forward pass of DCNet and CNFs, and the number of patients to evaluate. Also, the exact complexity of the direct optimization depends on the solver cost $\sigma$, it is at least $O(n \times k \times \sigma)$. This becomes costly for large-scale datasets. In contrast, our policy network is trained only once. Hence, upon deployment, it is constant (i.e., $O(1 \times \sigma)$ when measured in terms of $\sigma$).
> - The argmax cannot be determined analytically, because it depends on the learned nuisance parameters. However, when comparing our predicted policy against directly optimized dosaging, we observed barely any differences. One reason could be that the policy network introduces additional regularization for observations in low overlap regions.
> ## Response to “Minor”
> Thank you! We will add the suggested changes to our paper.
> “Filtering for T (L289)”: MIMIC-IV is a large dataset of patients in intensive care units for different causes, where we filter for patients who are ventilated.
> ## Response to “Limitations”
> Thank you! We will expand our discussion around further potential failure scenarios and possible downstream impacts when applying our method in clinal practice.

---

> > ### Comment · Reviewer_aVCP · 2023-08-14
> >
> > Thank you authors for the response, it has resolved most of my questions. I'm happy to raise my overall rating from 5 to 6.
> >
> > Feedback on the response:
> > - Thank you for clarifying my "weak overlap" question, sorry I missed that.
> > - Please include the compute complexity analysis (e.g., a short paragraph in appendix) to support the claim that policy network is more efficient at inference.
> >
> > I have some further clarifications:
> > - **PDF Table 2**: it seems that "MLP+reliable" led to competitive performance on MIMIC but not TCGA.
> >   - What's different about the two datasets? The response mentions "highdimensional confounders" but I'm still a little confused. Can you provide a small table summarizing the characteristics and dimensions? Currently it is hard to locate this information, I see MIMIC dosages are respiratory rate and tidal volume, but I can't seem to find what the dosages are for TCGA.
> >   - This indicates perhaps the proposed approach is not a one-size-fits-all solution. When might one consider using the simpler MLP as opposed to DCNet?
> > - Re **Response to Limitations**: I suggest the authors elaborate what specific limitations they plan to discuss, e.g. how might proposed approach fail, when might it not be the best approach.
> >   - One of the questions by Reviewer 5Gdr is "what is the harm of assuming independent dosage dimensions". I think this is very problem dependent. For example, [Tang et al. 2022](https://openreview.net/forum?id=Jd70afzIvJ4) found that for certain problems it is beneficial to assume independent treatments, as it allows for extrapolation into "underexplored regions" aka regions with "limited overlap". This includes not only when the treatments/dosages are truly independent, but also when there is positive interaction (as opposed to negative interaction, e.g. adverse drug-drug interaction). More discussions about this and what assumptions or domain knowledge is available for the applications of interest should help better clarify the paper contributions.
> >
> >     Tang et al. "Leveraging Factored Action Spaces for Efficient Offline Reinforcement Learning in Healthcare". NeurIPS 2022.
> >
> > I'm keeping my confidence score at 3, given my somewhat incomplete expertise in causal inference (cf. other reviewers), but personally I believe this paper does bring some good contributions.

---

> > > ### Author Response · Authors · 2023-08-15
> > > **1/2: Discussing Limitations**
> > >
> > > Thank you very much for your response and for updating your overall rating!
> > >
> > > As you suggested, we are going to include the complexity analysis of our previous response in the Appendix of our accepted paper to further support the efficiency of our method at inference time compared to direct optimization.
> > >
> > > We are happy to respond to your further questions:
> > > - **PDF Table 2:**
> > >     - Please find the table summarizing the characteristics and dimensions of our standard setting below. Please note that we are in a semi-synthetic data setting (see also **Appendix B**) so that we can vary the dimension of modeled dosages of $T$ in our ablation studies (as in **Main Paper, Fig. 3**;  **Appendix, Fig. 2**;  **Response PDF, Fig. 2**). Here, we can give meaning to the additional dosages by additional parameters to control for mechanical ventilation, and additional drugs for chemotherapy, respectively.
> > >
> > > |            |  MIMIC | |   TCGA ||
> > > | :---------------- | ------: | :----| ------: | :----|
> > > | Variable | Dim. | Meaning  | Dim.| Meaning |
> > > | $X$      |      33 | patient covariates (age, sex, respiratory and cardiac measurements) | 4000 |  gene expression measurements |
> > > $T$ |  2 | respiratory rate, tidal volume | 2 | chemotherapy drugs (e.g., doxorubicin, cyclophosphamide) |
> > > | $Y$ | 1  |chance of patient survival  | 1  |chance of patient survival  |
> > >
> > > - It has been shown that using a naïve neural network with $T$ and  $X$ concatenated as inputs (called “MLP” in our experiments) can lead to diminishing effect estimates of $T$ when $X$ is high-dimensional (see reference Shalit et al. 2017). Instead, our DCNet enforces the expressiveness of the effects of the dosage combinations by its custom prediction head and, therefore, is highly powerful for highdimensional settings (as in TCGA with 4000-dimensional features $X$). Intuition is also provided in **Appendix D**. In lower dimensions (as in MIMIC with 33-dimensional features), this benefit is simply smaller, and the MLP shows competitive performance wrt. to the selected policy. Nevertheless, when evaluating with multiple runs over different train/val/test splits, DCNet also shows better performance than MLP at MIMIC, especially wrt. to the variation (**Response PDF, Table 3&4**). Hence, our DCNet should also be preferred in lower-dimensional settings.
> > > In contrast to that, a possible setup where the MLP could outperform DCNet is in settings with extremely unsmooth, stepwise dose-response surfaces. While this setup is highly unlikely for dosing problems in medicine, we could imagine this might be the case when applying our method to different modalities, e.g., personalized advertisement in marketing.
> > > - **Response to limitations**:
> > >     - Thank you for the additional input regarding the potential harms of assuming independence between the dosages. While we show the advantages of modeling dependencies in our setting empirically (see our response **Additional evidence for modeling drug-drug interactions** to Reviewer 5Gdr), we would also like to state the importance from a perspective using domain knowledge. As also implied in Tang et al. 2022, while modeling independence can allow for better extrapolation in regions with limited overlap in the case of positive interactions, this does **not hold for non-monotonous or negative (adverse) interactions**. Especially in chemotherapy, negative drug-drug interactions occur frequently (e.g., a combination of doxorubicin and cyclophosphamide can increase the risk of heart problems (Ismail et al. 2020)). Hence, even if it might result in limited extrapolation, for following the Hippocratic Oath ("First do no harm") **modeling potential interactions is fundamental for reliable decision-making in medicine**.
> > >
> > >  To incorporate your feedback, we would like to present our first draft of our revised **Limitations** section in the **following comment:**
> > >
> > > References: \
> > > Shalit, Uri, Fredrik D. Johansson, and David Sontag. "Estimating individual treatment effect: generalization bounds and algorithms." International conference on machine learning. PMLR, 2017. \
> > > Ismail, Mohammad, et al. "Prevalence and significance of potential drug-drug interactions among cancer patients receiving chemotherapy." BMC cancer 20 (2020): 1-9. \
> > > Tang et al. "Leveraging Factored Action Spaces for Efficient Offline Reinforcement Learning in Healthcare". NeurIPS 2022.

---

> > > > ### Author Response · Authors · 2023-08-15
> > > > **2/2: Discussing Limitations**
> > > >
> > > > > **Limitations:** Our method makes an important contribution over existing literature in personalized treatment design for dosage combinations by adjusting for the naturally occurring but non-trivial problems of drug-drug interactions and limited overlap. However, the complexity of real-world data in medical practice can limit the applicability of our method in certain ways. (i) Our method does not account for the ignorability assumption, which can result in biased estimates in the case of unobserved confounders or other missing data. (ii) We apply our method in a static treatment setting, whereas, in several clinical applications, time-series data is available, e.g., sequences of varying dosages, multiple treatment cycles, and right-censored data. Therefore, future work may consider extending our method to handle additional sources of bias and violation of assumptions, and even more complex data settings. Furthermore, our method is developed for reliable, high-stakes decision-making in medicine. When transferred to different applications, however, other methods may be preferred. For instance, our method relies on the — for medical applications reasonable —  assumption of smooth dose-response surfaces. In settings with extremely unsmooth, stepwise dose-response surfaces, tree-based methods, or an MLP may be more suitable for dose-response estimation. Also, our method aims at reliable policy learning, avoiding areas with limited overlap to minimize potential harm. In different applications such as marketing with low risk at the downside, in contrast, one might even want to explicitly target these unreliable regions to maximize the probability of acquiring a new customer and hence would prefer to apply a different method.
> > > >
> > > > We hope this answers your questions fully and satisfactorily, especially by providing clear actions on how we present our revised paper. We hope that you can recommend acceptance of our paper.

---

### Official Review · Reviewer_bKrg · 2023-06-29

**Soundness:** 3 good
**Presentation:** 4 excellent
**Contribution:** 3 good
**Rating:** 7
**Confidence:** 4

**Summary:**

The paper deals with an important and interesting problem in personalized decision making. In particular, the paper considers the setting when there are multiple continuous treatments available and models the joint effect of dosage combinations. A novel off-policy learning algorithm is proposed to solve the problem using the well-constructed network architecture. Specifically, the paper also considers the issue when the (strong) positivity assumption of the propensity score does not hold. To solve this problem, the paper modifies the proposed algorithm to achieve a reliable estimation of the policy value by avoiding regions with limited overlap. Finally, several experiments and real data analysis are provided to demonstrate the performance of the proposed method.

**Strengths:**

1. The paper has a good presentation to illustrate the target problem, underlining causal framework, proposed methods, and related advantages. The authors have skillfully laid out the paper, providing a clear and concise overview of the problem at hand. The methods proposed by the authors are well-described and thoroughly explained, ensuring that readers can easily grasp the technical aspects of the approach. Furthermore, the paper showcases the advantages of the proposed methods, demonstrating how they address the limitations of existing approaches and offer novel contributions to the field. The well-designed format of presenting the problem, framework, methods, and advantages in a cohesive and coherent manner enhances the overall quality of the paper.

2. The paper provides real data analysis in multiple dosage settings. This empirical analysis adds substantial value to the research as it validates the proposed methods and demonstrates their effectiveness in practical scenarios. Moreover, the multiple dosage settings explored in the analysis indicate the robustness of the proposed methods, making the findings applicable to a wider range of scenarios.

**Weaknesses:**

1. The paper addresses a specific issue related to limited overlap, which can lead to unreliable estimations of policy value. However, it is worth noting that similar and related problems have been discussed in the literature. To the best of our knowledge, the following papers have explored these problems:

(1). Ma et al. (2023) Learning Optimal Group-structured Individualized Treatment Rules with Many Treatments. Journal of Machine Learning Research.

(2). Ma et al. (2022) Learning Individualized Treatment Rules with Many Treatments: A Supervised Clustering Approach Using Adaptive Fusion. NeurIPS.

Although the above papers primarily focus on settings involving discrete treatments, they have also carefully analyzed the limited overlap problem. Specifically, they examine cases where the number of treatments approaches $+\infty$, which can be treated as one of the special cases of continuous treatment, and propose methods for identifying latent treatment grouping structures (combine treatments or combine levels of doses) to address issues of unbalanced treatment assignment and limited overlap. It would be beneficial to include further discussion and comparisons with these methods in the paper, as they may offer valuable insights and contribute to the overall understanding of the problem.

2. The paper mentions an important term, $\bar{\epsilon}$, which acts as a pre-specified threshold controlling the minimum overlap required for the policy estimation when shrinking the treatment-covariate space. It is interesting and important to investigate and provide guidelines for selecting an appropriate value for $\bar{\epsilon}$ in the experimental analysis. This could involve sensitivity analysis or exploring different threshold values to evaluate their impact on the estimated policies and their performance metrics. Assessing the robustness of the proposed method to the choice of $\bar{\epsilon}$ would provide insights into its practical applicability and generalizability across various scenarios.

**Questions:**

1. In contrast to the approaches mentioned in the provided references, which focus on combining similar treatments and estimating treatment structures, this paper tackles the limited overlap problem in the context of continuous dosage by constraining the search space and shrinking the entire searching space. While the methods may differ in their specific techniques, there could be interesting connections between the two approaches. Exploring and discussing these potential connections could provide valuable insights into how different strategies can address the limited overlap problem across different treatment settings. Such a discussion would contribute to a broader understanding of the problem. In particular, are there any connections between these two methods? That would be interesting to discuss.

2. When reducing the searching space to address the limited overlap problem, there may be concerns about whether the estimated policy with the proposed method leads to sub-optimal values at the population level. Understanding the potential trade-offs between reducing the search space and achieving optimal population-level outcomes may be valuable for both theoretical considerations and practical applications. If reducing the searching space, will the estimated policy with proposed method leads to a sub-optimal value in the population level?



**Limitations:**

Please see it in Weaknesses section.

---

> ### Author Rebuttal · Authors · 2023-08-09
>
> Thank you for your detailed, constructive, and positive feedback!
>
> ### Response to “Weaknesses”
> 1. **Importance of continuous vs. discretized treatments.** Thank you for introducing this very interesting and related additional literature to us!\
> We understand the importance of benchmarking our framework for continuous treatments against methods that work instead on discretized settings. In principle, translating our continuous setting into a discretized setting would allow us to use a variety of established methods (such as the works that you cited above). We thus added experiments where we discretize the setting and then evaluate the solution quality (see **Response PDF, Table 4&5**). We find that the discretized approaches are consistently inferior compared to our approach which directly operates on continuous values. In particular, we also benchmark an “oracle” discretized baseline (where we assume perfect knowledge of the dose-response function and perfect individual treatment assignment). This serves as an upper bound for all policy learning methods with discretization. Still, that discretization leads to a higher regret than our proposed method, which confirms the advantages of leveraging continuous dosage information. \
> **Action:** We will cite the papers. We will add further discussions and comparisons between our method and the proposed papers on learning individualized treatment rules with many treatments. Specifically, we will discuss the connections and differences of the methods for tackling the challenges of limited overlap and biased treatment assignment, as well as the role of discrete and continuous treatment. We will also add the numerical results from the discretization baselines to our revised paper.
>
> 2. **Guidelines to select the reliability threshold.** We thank the authors for raising the important point of how to select the reliability threshold $\overline{\varepsilon}$ and for giving us the opportunity to elaborate on the approach we used in our work. We offer a detailed description in **Appendix C** and summarize key points in the following. For our framework, we developed a heuristic guideline to choose $\overline{\varepsilon}$ dependent on $x\%$-quantiles of the estimated GPS on the train dataset. As a result, we aim to learn optimal dosages, which have at least the same estimated reliability as $x\%$ of the observed data. In our experiments, we set $x=5$. Furthermore, we performed a series of sensitivity analysis where we explore different threshold values and then evaluate the impact of $\overline{\varepsilon}$ on the regret of the estimated policies. We show the results in **Response PDF, Figure (1)**. We find that, depending on the dataset, different quantiles for selecting $\overline{\varepsilon}$ can lead to small performance differences. However, we find that lower values tend to lead to less variability. Also, we find that, even when there is higher variability within the $k$ runs, for every setting a run with low regret is selected. This demonstrates that, by using our proposed guideline, our method yields **robust results also over different values of the reliability threshold**.
>
> ### Responses to “Questions”
> 1. We fully agree that discussing potential connections between the provided references and our method will contribute to our paper in terms of providing an even more extensive background of reliable off-policy learning under complex treatments with limited overlap. \
> **Action:** We will cite the papers and add a discussion to our revised paper. We will also benchmark the original methods and our “oracle” approach from above to show the importance of dealing with continuous treatments in medicine.
>
> 2. We thank you for the interesting question and the opportunity to discuss the effects of reducing the search space on the population-level outcomes. Importantly, even though our framework reduces the search space, it does so in a way that we still optimize against the optimal solution.
> - **Theoretical explanation.** We would like to emphasize that our method reduces the search space of the treatments during policy learning on an individualized level (conditioned on the patient covariates), but not necessarily on the population-level. This ensures that the treatment recommendation is still flexible across different patients and thus enables optimal individualized policy learning. Formally, recall that the empirical policy value $\hat{V}(\pi) = \frac{1}{n}\sum_{i=1}^{n} \hat{\mu} \left( \pi(x_i), x_i \right)$ is a metric for the population-level performance of a policy. It is then the same as the mean of the individual predicted outcomes. Hence, optimizing over the individual level and the population level leads to the same optimal policy. Hence, our framework which constrains the search space still targets the optimal policy at the population level.
> - **Empircial confirmation.** We also demonstrate the above empirically in our experiments. Here, we measure the regret which is the difference between the policy value of the optimal policy and the true policy value of our learned policy. We observe that our framework constantly outperforms the policies with unrestricted search spaces (called “naive” in our paper) wrt. to the regret. As a result, this provides empirical evidence that demonstrates the strong performance of our framework at a population level.

---

> > ### Comment · Reviewer_bKrg · 2023-08-14
> >
> > Thanks for the authors providing the clarifications for my review. That would be good if the paper can add the comparison about treatment combination between discrete and continuous treatments. I have raised my score.

---

> > > ### Author Response · Authors · 2023-08-15
> > >
> > > Thank you very much for your response and for increasing your overall rating! We will include all suggested points in the paper. Should you have any further questions or comments that should be addressed before a final decision, feel free to further post them here.

---

### Official Review · Reviewer_LeHo · 2023-07-06

**Soundness:** 3 good
**Presentation:** 3 good
**Contribution:** 2 fair
**Rating:** 5
**Confidence:** 4

**Summary:**

The paper introduces a new methodology to derive a policy to assign individualized dosage combinations relying on retrospectively collected observational data. The method relies on off-policy learning and try to account for the joint effect of multiple dependent dosages. A neural network is used to estimate individualized dose-response functions, accounting for the joint effect of multiple dependent dosages. The policy is then trained, avoiding regions with limited overlap to ensure reliable policy evaluation.

The developed method is then evaluated using semi-synthetic data relying on MIMIC-IV and TCGA. The robustness of the methodology is also assessed testing different levels of overlap for patients under different treatment regimens. The policy derived using the proposed method achieves low regret and low variance on all the experiments performed.

**Strengths:**

The paper combines the idea of several recently published papers to derive the global method accounting for joint effects of dosage combinations.
- A tensor product basis is used to estimate interaction effect and integrated into a model similar to [VCNet](https://arxiv.org/abs/2103.07861);
- The generalized propensity score is estimated using conditional normalized flows, increasing reliability and avoiding discretization of the problem;
- A NN is used for policy learning relying on the estimated dose response function and GPS;

A nice protocol is then performed, assessing the performance of the model in the presence of dose combinations with a joint effect.


**Weaknesses:**

In the optimization problem of equation (6) (line 237),  the objective function and constraints can be non convex. There is no reason when applying the method of Lagrangian multipliers to have a function that is concave in the Lagrangian multipliers $\lambda$. The method you rely on, [gradient descent ascent](https://arxiv.org/pdf/1906.00331.pdf) explicitly requires the concavity in $\lambda$. This can lead to sub optimality and all the nice properties of the method are lost.

Given this, we can expect that the methodology will induce a lot of variability. If part of it is captured with the random restart proposed, we still expected it to be highly unstable. The results don't reflect that with a surprising std at 0 for the policy derived, whatever the task is.
This might  be due to the overly simple ground truth dose response function and GPS. Elements to assess this hypothesis are provided in the Questions. This also might be due to seed tuning. It would be interesting to repeat the split into train, validation and test sets and take the variation in the best underlying regret into account.

**Questions:**

As mentioned above, the results are only displayed for one split, being potentially prone to seed tuning, it would be interesting to repeat the split into train, validation and test sets and take this variation into account.

Given that you have access to the ground truth for the dose response function and GPS, could you also provide the error to estimate them in your experiments. This could provide insights on why the observed std is a 0 and motivate a more realistic dose-response function.

In the TCGA dataset and MIMIC-IV, the outcome Y is taken as the patient's survival. It's very uncommon to get a dataset with full observation of Y and the outcome is often right censored. Did you think of a way to take this into account?

**Limitations:**

T is the assigned p-dimensional dosage combination. In practice most of the time it's a sequence of treatment, varying dosage and number of cycles. The setting in the paper is an advancement over even more simple settings but is far from reflecting the data you can access in clinical routine.

---

> ### Author Rebuttal · Authors · 2023-08-10
>
> Answer:
> Thank you for your detailed and constructive feedback. We improved our paper in the following ways:
>
> ## Responses to “Weaknesses”
> - Thank you for giving us the opportunity to clarify the optimization part in our paper. We agree that our final objective in Eq. (7) is non-concave in the neural network parameters $\theta$. However, for fixed $\theta$, Eq. (7) is a linear function in terms of the Lagrange multiplier $\lambda$ and thus **concave**. We therefore fulfill the assumption of the mentioned paper on gradient descent-ascent (Lin et al. 2021). Moreover, **we follow established literature** (e.g., Bui et al. 2022) on adversarial learning that uses gradient descent-ascent for non-convex concave objectives. The fact that Eq. (7) is non-convex in $\theta$ means that we may converge to local minima, yet which must not be attributed to the optimization but to the neural network approach. In our paper, we thus address this by re-running the optimization with different initializations of the learnable parameters and find that the performance remains stable (next bullet item).
>
> - We thank you for the opportunity to explain why the performance of our framework method between the $k$ different re-runs is stable and of low variability. To improve our paper, we have thus added new results: (1) We show the performance across different splits and find that there is low variability ( **PDF, Table 3**). (2) We added an additional semi-synthetic experiment with a more complex setting. Therein, we again find that the performance is stable and of low variability ( **PDF, Table 4**).
> To this end, our performance is **not** the result of seed tuning or overly simplistic nuisance functions. Rather, we exploit the properties of our medical setting (see Appendix B), as we have a complex multimodal dose-response function, where the dosage assignment is already informed to a certain degree by domain knowledge but must then be further optimized. This means, with increasing dosage bias $\alpha$, the probability mass around the optimal dosage combinations increases, while, in areas further away from the optimum, overlap becomes more limited. This is a reasonable assumption in medicine, as clinical practice is based on expert knowledge for deploying drug-dosage prescriptions, which minimizes the risk of observing harmful treatment assignments frequently.
>
> **Action:** We will add additional numerical results that demonstrate that our performance is stable to our paper. As shown above, the results confirm the effectiveness of our framework.
>
> ## Responses to “Questions”
> We thank the reviewer for the suggestions and added the following experiments.
>
> - **Performance across different splits (PDF,  Table 3&4)**: We find that our novel method consistently outperforms the baselines, yielding a consistently lower regret and lower variability. Note that this is *between* the different seeds as well as *within* the seeds between the k=5 restarts. As such, we conclude that the performance of our framework is robust.
>
> - **Results for nuisance estimators (PDF, Table 1)**: We further report the performance of our nuisance estimators. Here, for the dose-response estimators we show: (1) the performance on the test dataset measured by the mean integrated squared error (MISE) on the whole covariate treatment space; and (2) the MISE only in regions with sufficient overlap. Our results demonstrate that our DCNet is **highly effective** in learning the dose-response function. Compared to the baselines, we observe that our DCNet does not try to learn the whole complex dose-response surface (which includes low-trust regions) but focuses strategically on the areas with sufficient overlap (as desired). This is key for our framework as it ensures accurate targeting of the optimal policy. We provide further intuition in Appendix D.\
> We also report results from a more complex setting **(PDF, Table 4)** to show that our method generalizes to other complex scenarios. For this, we model the GPS as a multimodal conditional density and ensure that the optimal dosage combination is not necessarily located in regions around the maximal density. Importantly, our framework outperforms the baselines by a clear margin (i.e., lower regret and lower variability). This confirms the robustness of our framework.
>
> - **Extensions of our experimental setup.** Thank you for this important question. We designed our analysis around patient survival as it is often used for benchmarking in medical practice. Needless to say, our method can be adapted to other metrics of interest and even to right-censored data. Here, an interesting direction is to extend our framework with, for example, the approach in Curth et al. (2021). Importantly, our framework consists of three steps (dose-response estimation, GPS estimation, constrained optimization) which are not restricted to our specific setting of observed outcomes but are general. In particular, our framework can thus be extended to survival analysis with right-censored data in a straightforward manner.
>
> **Action:** We will add the above experiments to our revised paper. We will also add a discussion on how to handle right-censored data.
>
> ## Response to Limitations
> We agree that medical data often is even more complex than in our scenario, and we will add this to our discussion of limitations. Nevertheless, we would like to emphasize that our paper makes important contributions over existing literature in order to advance personalized treatment design in real-world clinical applications.
>
> ## Remark to Contribution
> Since the contribution was evaluated as “1 Poor”, we would like to emphasize the novelty of our work and refer to our General Author Rebuttal.
>
> Bui, Tuan Anh, et al. "A unified wasserstein distributional robustness framework for adversarial training." arXiv (2022)\
> Curth, Alicia,et. al. "Survite: Learning heterogeneous treatment effects from time-to-event data." NeurIPS (2021)

---

> > ### Comment · Reviewer_LeHo · 2023-08-16
> >
> > Thanks a lot for the answers and the additional experiments you performed.
> > My main concern was to observe an std at 0 for the policy derived, whatever the task is. Table 3 in the new pdf seems to address this. It's quite unusual however to have a confidence interval for the std and the std reported two time (in the mean column and in the std column). Maybe it would be more clear to just compute the mean and the std over the 5 runs and the 5 restarts (over the 25 results).
> > Given the new experiments, extending to more complex setup and better assessing the robustness of the global framework, I updated my overall rating to 5.

---

> > > ### Author Response · Authors · 2023-08-16
> > > **Interpretation of Response PDF, Table 3&4**
> > >
> > > Thank you for your response and for raising your overall rating! We are happy we could address your concerns sufficiently.
> > >
> > > We agree that **Response PDF, Table 3&4** may seem unusual at first sight. Here, want to elaborate on the advantages of our presentation and give a short intuition.
> > > 1. The column “**selected**" displays the final performance of our method when using the finally recommended policy (chosen by our validation loss in **Main Paper, Eq. 9**). Hence, its $ mean \pm sd $ can be considered the expected regret with Monte-Carlo-CV confidence intervals (including retraining the nuisance functions) and should be primarily used for evaluating the performance and for comparison with different policy learning methods. We observe that our method significantly outperforms the baselines.
> > > 2. The column “**mean**” represents the means within the $k$ restarts of the policy learning (step 3 in our method). As such, its values can indicate how a randomly selected policy out of the $k$ restarts would perform compared to the “selected one” (e.g., we observe that for the “naive” baselines, by using a naive MSE-loss selection often even suboptimal methods are selected, whereas our method shows robust behavior).
> > > 3. The column “**std**” shows the standard deviation within the $k$ restarts of the policy learning (step 3 in our method). As such, its values indicate how often similar performing policies are learned. It thus refers to the robustness within the $k$ runs. As a consequence, lower values imply that  (i) probably fewer restarts $k$ are needed to find the targeted optimum, and (ii) fewer local optima are learned, which reduces the risk of selecting a suboptimal policy.
> > >
> > > Hence, in combination with summarizing multiple re-runs over different splits, these three metrics give valuable insights into the robustness of our method. In contrast, when we would just calculate the mean of the different selected runs (or the mean of the means) and display the standard deviations over all 25 restarts, this would lead to incorrectly very wide confidence intervals, especially in very complex data settings where there might be a high standard deviation within the $k$ policy restarts but still a low variation between the finally selected policies of the split-re-runs. Hence, we suggest using the “**selected**” column ($ mean \pm sd $) for comparing the performance to all baselines methods (e.g., including the ones using discretization); and the other two columns (“**mean**”, “**std”**) to additionally show the inner robustness of our method compared to ablation baselines.
> > >
> > >
> > > We hope we could address all your concerns adequately and that you can recommend acceptance of our paper. If you still have any open questions or further suggestions on how we could improve our paper, we would be happy to address all of them during the remaining discussion period.

---

### Author Rebuttal · Authors · 2023-08-10

Thank you very much for the constructive and positive evaluation of our paper and your helpful comments! We addressed all of them in the comments below and uploaded additional results as a PDF file.

Our main improvements are the following:
- We provide further **extensive experiments**: \
(1) We report the performance of our nuisance estimators and show our framework is superior. \
(2) We report results for multiple train/val/test splits for (a) our existing setting and (b) for an even more complex, multimodal setting. Here, we demonstrate that the performance of our framework is robust across all settings. \
(3) We include additional causal baselines for (a) dose-response estimation, (b) GPS estimation, and (c) policy learning. We observe that all components of our framework lead to performance gains. \
(4) We show that our framework for continuous treatments outperforms baselines based on discretized treatments. \
(5) We compare additional, standard causal baselines and find that our framework still perfoms best. \
(6) We demonstrate the scalability of our framework to higher dimensions.

- We discussed the **connections of our method to further related work** (e.g., offline reinforcement learning, causal inference for discrete treatments) and clarify the necessity and advantages of using our method for reliable off-policy learning for dosage combinations.
- We elaborate on the **role of the reliability threshold** $\overline\varepsilon$ for reliable policy learning in our method. We further propose a guideline for selecting $\overline\varepsilon$. We also show that our framework is robust to different choices of $\overline\varepsilon$.

As a summary, we would like to emphasize our contribution again: To the best of our knowledge, we are the first to tackle the problem of learning optimal individualized dosage combinations from a causal perspective. For this, we adjust for the naturally occurring but non-trivial problems of drug-drug interactions and limited overlap. Crucially, our study setting is highly relevant in medical practice and present the foundation for personalized decision-making in cancer therapy and critical care.  Even more, we implemented additional baselines and show that our method clearly outperforms existing methods. Rather, our problem requires a tailored and non-trivial framework as ours. As such, we are confident to provide a valuable contribution toward learning reliable policies for dosage combinations in medicine.

We will incorporate all changes (labeled with **Action**) into the camera-ready version of our paper. Given these improvements, we are confident that our paper will be a valuable contribution to the machine learning for healthcare literature and a good fit for NeurIPS 2023.

---

### Decision · Program_Chairs · 2023-09-21

**Decision:**

Accept (poster)

**Comment:**

This paper proposes off-policy learning for drug-drug combinations. The method trains treatment specific response functions followed by policy learning that restricts to learning under conditions of overlap to extrapolate drug-drug combination effects. Experiments in TCGA and MIMIC on improving survival are demonstrated.

Overall the reviewers find the contribution valuable and the authors have made a significant effort in addressing clarifying questions and concerns from reviewers. I do believe all additional analysis should be incorporated in the main paper. Particularly the sensitivity analysis.

Otherwise I do not have concerns about acceptance.